# Genome-Wide Identification of 2-Oxoglutarate and Fe (II)-Dependent Dioxygenase (2ODD-C) Family Genes and Expression Profiles under Different Abiotic Stresses in *Camellia sinensis* (L.)

**DOI:** 10.3390/plants12061302

**Published:** 2023-03-14

**Authors:** Jingxue Han, Xiaojing Wang, Suzhen Niu

**Affiliations:** College of Tea Science, Guizhou University, Guiyang 550025, China

**Keywords:** *C. sinensis*, *Cs2ODD-C* genes, phylogenetic analysis, expression profile, abiotic stresses

## Abstract

The 2-oxoglutarate and Fe (II)-dependent dioxygenase (2ODD-C) family of 2-oxoglutarate-dependent dioxygenases potentially participates in the biosynthesis of various metabolites under various abiotic stresses. However, there is scarce information on the expression profiles and roles of *2ODD-C* genes in *Camellia sinensis*. We identified 153 *Cs2ODD-C* genes from *C. sinensis*, and they were distributed unevenly on 15 chromosomes. According to the phylogenetic tree topology, these genes were divided into 21 groups distinguished by conserved motifs and an intron/exon structure. Gene-duplication analyses revealed that 75 *Cs2ODD-C* genes were expanded and retained after WGD/segmental and tandem duplications. The expression profiles of *Cs2ODD-C* genes were explored under methyl jasmonate (MeJA), polyethylene glycol (PEG), and salt (NaCl) stress treatments. The expression analysis showed that 14, 13, and 49 *Cs2ODD-C* genes displayed the same expression pattern under MeJA and PEG treatments, MeJA and NaCl treatments, and PEG and NaCl treatments, respectively. A further analysis showed that two genes, *Cs2ODD-C36* and *Cs2ODD-C21*, were significantly upregulated and downregulated after MeJA, PEG, and NaCl treatments, indicating that these two genes played positive and negative roles in enhancing the multi-stress tolerance. These results provide candidate genes for the use of genetic engineering technology to modify plants by enhancing multi-stress tolerance to promote phytoremediation efficiency.

## 1. Introduction

Plants possess several mechanisms to cope directly with various abiotic stresses, such as extreme temperature, salt, and drought, owing to their sessile nature [1]. One of the important mechanisms for plants under environmental challenges is the regulation of secondary metabolites [2]. Plants can produce more than 200,000 phytochemicals through various metabolic enzymes [3,4]. In particular, oxygenation/hydroxylation reactions are very important for plant growth and development, which are catalyzed by the 2-oxoglutarate-dependent dioxygenase (2OGD) [5,6]. 2OGDs, the second-largest enzyme family in plants, require 2-oxoglutarate (2OG) and molecular oxygen as co-substrates, and ferrous iron Fe(II) as a cofactor to catalyze the oxidation of a substrate with concomitant decarboxylation of 2OG to form succinate and carbon dioxide. 2OGDs were involved in the biosynthesis of many secondary metabolites [6,7,8,9].

In *Arabidopsis*, the 2OGD superfamily is divided into three subfamilies: DOXA, DOXB, and DOXC, comprising 14, 14, and 96 *2OGD* genes, respectively. DOXA subfamily genes are mainly involved in the N-methyl group hydroxylation of certain genes [10], DOXB subfamily genes mainly participate in the hydroxylation of proline [11], and the DOXC subfamily, as the most functionally diverse protein subfamily, is associated with the biosynthesis of secondary metabolites, such as dioxygenases for auxin oxidation (DAOs), 1-minocyclopropane carboxylic acid oxidases (ACOs), and gibberellin (GA) 2-oxidases (GA2oxs, GA3oxs, and GA20oxs) for GA synthesis; and metabolism, flavanone 3-hydroxylases (F3Hs), anthocyanidin synthases (ANSs), and aromatic glucosinolates (AOPs) for glucosinolate synthesis [9]. The DOXC subfamily comprises 2-oxoglutarate and Fe-(II)-dependent dioxygenases (2ODDs), characterized by conserved 2OG-FeII_Oxy (PF03171) and DIOX_N (PF14226) domains [12]. Moreover, a phylogenetic analysis revealed that the DOXC subfamily proteins in land plants clustered into 57 clades, with 23 clades comprising *Arabidopsis 2ODD* genes [7].

Numerous studies have reported that 2ODDs play important roles in the growth and development processes of plants and participate in the regulation of secondary metabolite biosynthesis under various stresses. For example, overexpression of the *CrCOMT* gene (encoding the caffeic acid *O*-methyltransferase) from *Carex rigescens* enhanced salt tolerance in transgenic *Arabidopsis.* The *TaACO1* gene from wheat (*Triticum aestivum*) also belongs to the 2ODD gene family, which conferred salt sensitivity when overexpressed in transgenic *Arabidopsis* [13]. The turnover of auxin can be manipulated under salt treatment through inducing the expression of *GH3* genes and inhibiting *DAO* genes [14]. The 2ODD family members AOP and GSL-OH were reported to increase drought and salt tolerance by regulating glucosinolate biosynthesis and metabolism [15]. Moreover, the T-DNA insertional mutations of *GSL-OH* in *Arabidopsis* prevented the accumulation of 2-hydroxybut-3-enyl glucosinolate, which decreased the insect resistance of the plants [16]. Overexpression of the DOXC subfamily *F3H* gene from *Camellia sinensis* and *Lycium chinense* in tobacco and *Arabidopsis* increased their salt and drought tolerance, respectively [17,18]. In addition, overexpression of the *SlF3H* gene could improve the cold tolerance of *Solanum lycopersicum* by regulating jasmonic acid accumulation [19]. Ectopic expression of the DOXC subfamily *DoFLS1* gene from *Dendrobium officinale* promoted flavonol accumulation and enhanced tolerance to abiotic stress in *Arabidopsis* [20]. Overexpression of the DOXC subfamily gene *StGA2ox1* in potato improved stress tolerance compared with that of wild-type plants, including salt, drought, and low-temperature tolerance [21]. 

Tea (*C. sinensis*) is the oldest natural, nonalcoholic, caffeine-containing beverage and benefits human health due to its wealth of secondary metabolites, including catechins, theanine, polysaccharides, caffeine, and volatiles [22,23,24]. Abiotic stresses considerably affect the yield, quality, and even life of *C. sinensis*, and such adverse environmental conditions may reduce the performance of the *C. sinensis* with reduced yield from 65% [25]. Although the functions of *2ODD-*C genes have been extensively investigated in multiple model organisms, these enzymes have not been systematically analyzed in *C. sinensis.* In our current study, a comprehensive analysis of the Cs2ODD-C gene family in *C. sinensis* was performed for phylogenetic evolution, gene structure, conserved motifs, chromosome location, gene duplication, and expression patterns in different organs and under different abiotic stress conditions. In addition, two candidate genes (*Cs2ODD-C36* and *Cs2ODD-C21*) might play both positive and negative roles in regulating the multi-stress tolerance. Our results provide new insight into the function of *Cs2ODD-C* genes in *C. sinensis* and establish a knowledge base for further genetic improvement of *C. sinensis*. Through this study, we hope to develop a novel molecular basis to improve the multi-stress tolerance of plants and promote the development of phytoremediation technology for multi-stress situations.

## 2. Results

### 2.1. Genome-Wide Identification of Cs2ODD-C Gene Family in C. sinensis

To identify the Cs2ODD-C family genes in *C. sinensis*, the full-length sequence alignments of the DIOX_N (PF14226) and 2OG-FeII_Oxy (PF03171) domains were downloaded from the Pfam database and were used as queries to search the *C. sinensis* proteome [26]. A total of 153 *Cs2ODD-C* genes were identified in *C. sinensis* genome, encoding proteins ranging from 162 aa (Cs2ODD-C129) to 422 aa (Cs2ODD-C104) in length, with an average of 334.4 aa. The molecular weight of Cs2ODD-C family proteins ranged from 17.53 kDa (Cs2ODD-C129) to 47.36 kDa (Cs2ODD104), and the isoelectric points ranged from 4.18 (Cs2ODD-C130) to 9.55 (Cs2ODD-C47). The prediction of the subcellular localization of Cs2ODD-C family proteins showed that all proteins were distributed in the cytoplasm (Appendix A). The detailed information of Cs2ODD-C family genes is provided in Appendix A, including chromosome location, number of introns, and the gene start and end sites. 

Based on the valuable information for genome annotation, 146 of the identified *Cs2ODD-C* genes were localized on 15 *C. sinensis* chromosomes, with the remaining 7 genes (*Cs2ODD-C1*, *Cs2ODD-C17*, *Cs2ODD-C49*, *Cs2ODD-C70*, *Cs2ODD-C74*, *Cs2ODD-C95*, and *Cs2ODD-C107*) unanchored to chromosomes. As shown in Figure 1, the congregate regions and number of *Cs2ODD-C* genes are irregular across the 15 chromosomes. Chromosome 9 harbored the highest number of *Cs2ODD-C* genes (n = 18), whereas Chromosome 6 had the smallest number of genes (n = 3). Moreover, the percentage of *Cs2ODD-C* genes per chromosome varied from 0.09% on Chromosome 6 to 0.65% on Chromosome 15 (Appendix A).

### 2.2. Phylogenetic Analysis of 2ODD-C Gene Family among Arabidopsis, O. sativa, and C. sinensis

To further explore the relationships among Cs2ODD-C family genes, an unrooted phylogenetic tree in *C. sinensis, O. sativa*, and *Arabidopsis* was constructed by PhyML 3.0 software with the maximum-likelihood (ML) method. The Cs2ODD-C family genes clustered into 21 groups (Groups I to XXI) based on the tree topology and the functions of identified *AtODD-C* genes in *Arabidopsis* (Figure 2; Table 1). The number of *Cs2ODD-C* genes in different groups was different. Group XIV, including 20 *Cs2ODD-C* genes, was the largest group, followed by Group VI (15 genes), Group VIII (14 genes), and Group IV (13 genes). Group XVIII contained only one *Cs2ODD-C* gene, making it the smallest group. In addition, Groups IV, VIII, X, XII, XIII, XIV, XVII, XX, and XXI contained more *Cs2ODD-C* genes in *C. sinensis* than in *Arabidopsis* and *O. sativa*. In addition, two groups (Groups VII and XV) harbored only *At2ODD-C* or *Cs2ODD-C* genes without corresponding *Os2ODD-C* genes in *O. sativa*, indicating that the number of 2*ODD-C* genes may have increased in *C. sinensis* and *Arabidopsis* or that *O. sativa* lost these 2*ODD-C* genes during evolution.

### 2.3. Gene Structure and Conserved Motifs of Cs2ODD-C Family Genes 

To further investigate the gene structure diversity and motif composition of *Cs2ODD-C* genes, the conserved motifs of Cs2ODD-C proteins were analyzed and visualized using the MEME suite. Fifteen putative conserved motifs were identified in Cs2ODD-C proteins (Figure 3a and Appendix A). Motifs 2, 3, and 4 corresponding to the DIOX_N domain, and Motifs 1, 5, and 10 corresponding to the 2OG-Fe II_Oxy domain were shared by nearly all Cs2ODD-C proteins. The distributions of some conserved motifs varied among the groups, with some groups having specific conserved motifs. Motif 12 was specifically distributed in all members of Group XVII, whereas Motif 15 was specifically distributed in Groups XII and XIV. The members of Groups I–VI and VII–XI commonly contained Motifs 11 and 14. These results indicated that the specific motifs appearing in different subgroups might be associated with specific functions.

The exon/intron structural patterns of *Cs2ODD-C* genes are shown in Figure 3b and Appendix A. The number of exons for the 153 *Cs2ODD-C* genes varied from 1 to 11. A majority (138 of 153) of the *Cs2ODD-C* genes have 1–4 introns, and the other 15 *Cs2ODD-C* genes have 9–11 introns. *Cs2ODD-C* genes in the same subgroups consistently showed the same numbers and lengths of introns/exons, indicating that they have the same intron/exon organization. For instance, all genes in Group XIII contained only one intron, whereas most members of Groups X–XII possessed two introns, and all members of Group XVII had more than ten introns. Thus, this analysis further verified the topology of the phylogenetic tree of Cs2ODD-C family genes. 

### 2.4. Cis-Regulatory Elements in the Cs2ODD-C Promoter Regions

*Cis*-regulatory elements are responsible for transcriptional regulation by binding to transcription factors. The *cis*-elements in the promoter regions of *Cs2ODD-C* genes were analyzed using PlantCARE online software (Appendix A). There were 23 functionally annotated *cis*-elements identified, and these were further classified into three categories: stress-responsive elements (MYB, MBS, AE-box, ABRE, TCA-motif, box-III, and TATA), hormone-responsive elements (RY-element, P-box, GA-motif, TATC-motif, ABRE, and Sp1), and light-responsive elements (AT-rich element, I-box, G-box, GA motif, AT1-motif, LTR, ATCT-motif, ACE, W-box, and TGA-box). The *Cs2ODD-C* genes contained numerous stress-, hormone-, and light-responsive elements, suggesting that these genes might be involved in response to light signaling, plant growth and development, stresses, and hormones. In addition, the promoters of *Cs2ODD-C36* and *Cs2ODD-C21* genes contained many stress-related *cis*-elements, such as CGTCA-motif, TATC-box, GARE-motif, ABRE, MBS, P-box, TCA-motif, and GACG-motif, indicating that these two genes might play crucial roles in response to various abiotic stresses in *C. sinensis* (Appendix A). 

### 2.5. Evolutionary Patterns of the Cs2ODD-C Gene Family

Gene duplication and divergence play important roles in the evolutionary momentum of the genome, and tandem duplication (TD) and whole-genome duplication (WGD) contribute to evolutionary novelty and genome complexity [27]. Among the 153 *Cs2ODD-C* genes, 36 (23.52%) and 39 (25.49%) were duplicated and retained from WGD/segmental duplication and TD, respectively (Appendix A), indicating that these two gene duplication events mainly contribute to the expansion of Cs2ODD-C family genes in *C. sinensis.*


A collinearity analysis of the Cs2ODD-C gene family in *C. sinensis* was conducted to further study the underlying evolutionary processes. A total of 33 *Cs2ODD-C* genes from 23 segmental duplication events were identified in *C. sinensis*, which accounted for 91.6% of WGD-type *Cs2ODD-C* genes (Figure 4a; Appendix A). *Cs2ODD-C* genes were located within synteny blocks on all chromosomes. In addition, the ratio of non-synonymous to synonymous substitutions (Ka/Ks) is a useful measure of the strength and mode of natural selection acting on protein-coding genes. A Ka/Ks ratio of 1 is indicative of neutral selection (no specific direction), lower than 1 indicates purifying selection, and higher than 1 indicates positive selection [28]. In the present study, the Ka/Ks ratio of 23 *Cs2ODD-C* gene pairs was less than one, implying that these genes are under negative purifying selection, which maintained the functions of the Cs2ODD-C gene family in *C. sinensis* (Appendix A). Moreover, Ks was usually used to estimate the evolutionary dates of genome or gene duplication events. Te WGD/segmental duplicated events in *C. sinensis* occurred from 0.91 (Ks = 0.0272) to 81.51 mya (Ks = 2.5541). 

The orthologous relationships of 2ODD-C family genes among *O. sativa*, *C. sinensis*, and *Arabidopsis* were further evaluated (Figure 4b,c; Appendix A). A total of 36 orthologous gene pairs of *2ODD-C* were identified between *C. sinensis* and *Arabidopsis*, whereas there were only 5 orthologous gene pairs of *2ODD-C* between *C. sinensis* and *O. sativa* identified in this study (Figure 4b,c). The much higher number of orthologous events of *Cs2ODD-C*–*At2ODD-C* than that of *Cs2ODD-C*–*Os2ODD-C* indicated that *C. sinensis* is more closely related to *Arabidopsis* than to *O. sativa.*

### 2.6. Expression of Cs2ODD-C Family Genes in Different Tissues of C. sinensis

The expression patterns of *Cs2ODD-C* genes in different tissues of *C. sinensis* were evaluated using the previous RNA-seq data from the Tea Plant Information Archive (TPIA) database [29]. Four *Cs2ODD-C* genes (*Cs2ODD-C40*, *Cs2ODD-C47*, *Cs2ODD-C53*, and *Cs2ODD-C58*) were not detected in any tissue of *C. sinensis.* The expression patterns of the remaining 149 *Cs2ODD-C* genes in different tissues of *C. sinensis* could be grouped into three types (Figure 5). Type I contained 30 *Cs2ODD-C* genes, which were constitutively expressed in all investigated organs of *C. sinensis*. Type II also contained 30 *Cs2ODD-C* genes, which displayed a tissue-specific expression pattern, including 25 *Cs2ODD-C* genes specifically expressed in the roots, 3 *Cs2ODD-C* genes (*Cs2ODD-C43*, *Cs2ODD-C79*, and *Cs2ODD-C129*) expressed in the mature leaves, and 2 genes (*Cs2ODD-C127* and *Cs2ODD-C128*) expressed in the stems. Type III included 24 *Cs2ODD-C* genes, 22 of which exhibited a high expression level in the vegetative organs, whereas the expression levels of the other 2 genes (*Cs2ODD-C22* and *Cs2ODD-C5*) were higher in the fruits and flowers. Interestingly, the expression patterns of some *Cs2ODD-C* genes in the same group were similar. The genes in Group III showed high expression levels in all sampled tissues. Of the 13 genes in Group IV, 10 showed significantly high expression levels in the roots. Sixteen genes (*Cs2ODD-C29*, *Cs2ODD-C31*, *Cs2ODD-C119*, *Cs2ODD-C56*, *Cs2ODD-C37*, *Cs2ODD-C38*, *Cs2ODD-C96*, *Cs2ODD-C109*, *Cs2ODD-C11*, *Cs2ODD-C44*, *Cs2ODD-C5*1, *Cs2ODD-C 113*, *Cs2ODD-C12*, *Cs2ODD-C42*, *Cs2ODD-C23*, *Cs2ODD-C25*, *Cs2ODD-C1*, *Cs2ODD-C133*, and *Cs2ODD-C147*) were not expressed in any of the investigated organs or tissues. 

### 2.7. Expression Pattern of Cs2ODD-C Family Genes under MeJA Treatment

The expression patterns of 153 Cs2ODD-C family genes in *C. sinensis* seedlings that experienced MeJA treatment were evaluated used previous RNA-seq data from the TPIA database (Figure 6). Under MeJA treatment, 11 *Cs2ODD-C* genes (*Cs2ODD-C49*, *Cs2ODD-C50*, *Cs2ODD-C36*, *Cs2ODD-C27*, *Cs2ODD-C110*, *Cs2ODD-C57*, *Cs2ODD-C8*, *Cs2ODD-C5*, *Cs2ODD-C153*, *Cs2ODD-C125*, and *Cs2ODD-C18*) were upregulated at both 24 h and 48 h, whereas 14 *Cs2ODD-C* genes (*Cs2ODD-C104*, *Cs2ODD-C54*, *Cs2ODD-C91*, *Cs2ODD-C39*, *Cs2ODD-C3, Cs2ODD-C87*, *Cs2ODD-C21*, *Cs2ODD-C34*, *Cs2ODD-C123*, *Cs2ODD-C146*, *Cs2ODD-C134*, *Cs2ODD-C145*, *Cs2ODD-C109*, and *Cs2ODD-C88*) showed downregulated expression at both time points (Figure 6), and this may indicate that these were long-term response genes in *C. sinensis* under MeJA exposure. Moreover, the expression levels of 47 *Cs2ODD-C* genes increased to a maximum at 24 h and then sharply decreased at 48 h of MeJA treatment, which may indicate that these were short-term response genes in *C. sinensis* under MeJA exposure. Both long-term and short-term response genes in *C. sinensis* might play important roles in the response to MeJA. In addition, 26 *Cs2ODD-C* genes showed only upregulated expression at 48 h after MeJA treatment; this may indicate that these genes are relatively less responsive to MeJA (Figure 6). However, there was no significant change in the expression levels of 20 *Cs2ODD-C* genes between the tissues treated with MeJA and untreated control tissues, indicating that these genes might not be involved in the response to MeJA in *C. sinensis* (Figure 6). 

### 2.8. Expression Pattern of Cs2ODD-C Family Genes under PEG Treatment

The expression levels of Cs2ODD-C family genes in *C. sinensis* seedlings under PEG treatment were evaluated using the previous RNA-seq data from the TPIA database (Figure 7). The expression levels of 28 *Cs2ODD-C* genes significantly increased at both 24 h and 48 h after PEG treatment, whereas 59 *Cs2ODD-C* genes showed downregulated expression at both time points. Moreover, the expression levels of 14 *Cs2ODD-C* genes (*Cs2ODD-C106*, *Cs2ODD-C49*, *Cs2ODD-C5*0, *Cs2ODD-C66*, *Cs2ODD-C27*, *Cs2ODD-C46*, *Cs2ODD-C4*, *Cs2ODD-C2*, *Cs2ODD-C107*, *Cs2ODD-C110*, *Cs2ODD-C153*, *Cs2ODD-C108*, *Cs2ODD-C136*, and *Cs2ODD-C42*) increased to a maximum at 24 h after PEG treatment and then decreased sharply at 48 h after PEG treatment, which may indicate short-term response genes in *C. sinensis* under PEG exposure. Both long-term and short-term response genes in *C. sinensis* might play important roles in the response to PEG. In addition, the expression of 13 *Cs2ODD-C* genes was only significantly upregulated at 48 h after PEG treatment, suggesting that these genes are relatively less responsive to PEG stress (Figure 7). However, there was no significant difference in the expression levels of 27 *Cs2ODD-C* genes between the PEG treatment and control groups, thus indicating that these genes might not be associated with drought stress in *C. sinensis*.

### 2.9. Expression Pattern of Cs2ODD-C Family Genes under NaCl Stress

Salinity can regulate the growth and development of plants, and salt stress can severely affect plant growth, thus reducing the plant yield. Therefore, we evaluated the expression levels of the 153 Cs2ODD-C family genes in *C. sinensis* seedlings that experienced NaCl treatment, using the previous RNA-seq data from the TPIA database. We found that the expression of 21 *Cs2ODD-C* genes was significantly upregulated at both 24 h and 48 h after NaCl treatment, whereas the expression of 58 *Cs2ODD-C* genes was significantly downregulated at both 24 h and 48 h after NaCl treatment, indicating that these genes may be long-term response genes in *C. sinensis* under salt stress (Figure 8). Moreover, the expression levels of 20 *Cs2ODD-C* genes increased to a maximum at 24 h and then decreased sharply at 48 h of NaCl treatment (Figure 8), and this may indicate that these are short-term response genes in *C. sinensis* under salt stress. In addition, 16 *Cs2ODD-C* genes were only significantly upregulated 48 h after NaCl treatment, implying that these genes are relatively less responsive to salt stress (Figure 8). However, the expression levels of 25 *Cs2ODD-C* genes showed no significant change after NaCl treatment in comparison with those of the controls, thus indicating that these genes might not be involved in the response to the salt stress of *C. sinensis* (Figure 8).

Pairwise comparisons of gene expression levels among MeJA, PEG, and NaCl treatments were performed to further evaluate the expression pattern of *Cs2ODD-C* genes in *C. sinensis*. The expression of three *Cs2ODD-C* genes (*Cs2ODD-C125*, *Cs2ODD-C36*, and *Cs2ODD-C8*) was significantly upregulated at both 24 h and 48 h after MeJA and PEG treatments, whereas four *Cs2ODD-C* genes (*Cs2ODD-C104*, *Cs2ODD-C109*, *Cs2ODD-C54*, and *Cs2ODD-C21*) showed downregulated expression at both time points. The expression levels of six *Cs2ODD-C* genes (*Cs2ODD-C2*, *Cs2ODD-C4*, *Cs2ODD-C42*, *Cs2ODD-C46*, *Cs2ODD-C66*, and *Cs2ODD-C107*) increased to a maximum at 24 h and then decreased at 48 h of both the MeJA and PEG treatments. Only one gene (*Cs2ODD-C93*) showed upregulated expression at 48 h after both the MeJA and PEG treatments (Figure 6 and Figure 7, and Appendix A). As shown in Figure 6 and Figure 8, the expression of two *Cs2ODD-C* genes (*Cs2ODD-C27* and *Cs2ODD-C36*) was upregulated at 24 h and 48 h of MeJA and NaCl treatments, whereas three *Cs2ODD-C* genes (*Cs2ODD-C109*, *Cs2ODD-C143*, and *Cs2ODD-C21*) showed downregulated expression at both time points in both treatments. The expression levels of four *Cs2ODD-C* genes (*Cs2ODD-C30*, *Cs2ODD-C4*, *Cs2ODD-C72*, and *Cs2ODD-C95*) increased to a maximum at 24 h and then decreased at 48 h of MeJA and NaCl treatments. Furthermore, four *Cs2ODD-C* genes (*Cs2ODD-C116*, *Cs2ODD-C117*, *Cs2ODD-C67*, and *Cs2ODD-C52*) were only significantly upregulated at 48 h of MeJA and NaCl treatments (Figure 6 and Figure 8; Appendix A). Under the PEG and NaCl treatments, nine *Cs2ODD-C* genes (*Cs2ODD-C117*, *Cs2ODD-C 140*, *Cs2ODD-C35*, *Cs2ODD-C36*, *Cs2ODD-C39*, *Cs2ODD-C44*, *Cs2ODD-C121*, and *Cs2ODD-C6*) showed significantly upregulated expression at 24 h and 48 h, while 37 *Cs2ODD-C* genes showed downregulated expression at both time points. The expression levels of two *Cs2ODD-C* genes (*Cs2ODD-C110* and *Cs2ODD-C48*) increased to a maximum at 24 h and then decreased at 48 h of PEG and NaCl treatments (Figure 7 and Figure 8; Appendix A). It is worth noting that one *Cs2ODD-C* gene (*Cs2ODD-C36*) showed upregulated expression at both 24 h and 48 h for the MeJA, PEG, and NaCl treatments, whereas the *Cs2ODD-C* gene (*Cs2ODD-C21*) was downregulated at both time points in all three treatments (Figure 6, Figure 7 and Figure 8; Appendix A), suggesting that these three genes might play important roles under various abiotic stresses in *C. sinensis*. 

### 2.10. Validation of Expression Patterns of 12 Cs2ODD-C Genes under PEG, NaCl, and MeJA Treatments Using qRT-PCR

To validate the reliability of RNA-seq results, twelve *Cs2ODD-C* genes with high expression after at least two treatments were selected to verify the expression patterns by reverse transcription–quantitative polymerase chain reaction (RT-qPCR) experiments (Figure 9). Expression comparisons were performed in *C. sinensis* seedlings treated with PEG, NaCl, and MeJA, and the expression trends in the RT-PCR results were in agreement with the RNA-Seq data.

### 2.11. Tertiary Structures of 12 Candidate Stress-Related Cs2ODD-C Proteins 

Based on the prediction from SWISS-MODEL (https://swissmodel.expasy.org, accessed on 15 March 2022), the protein tertiary structures of 12 candidate stress-related Cs2ODD-C proteins were constructed according to the C300xA [2ogFe(II) oxygenase family protein] template (Figure 10). The three highest scoring templates were 6lsv.2.A (JOX2), 6ku3.1.A (GA2ox3), and 5o7y.1.A (T6ODM). Seven Cs2ODD-C proteins (Cs2ODD-C27, Cs2ODD-C35, Cs2ODD-C36, Cs2ODD-C39, Cs2ODD-C72, Cs2ODD-C80, and Cs2ODD-C107) contained all three templates (6lsv.2.A, 6ku3.1.A, and 5o7y.1.A), which were used as short- and long-term response genes in *C. sinensis* under MeJA, PEG, and NaCl treatments. The 6lsv.2.A template was present in all 12 Cs2ODD-C proteins, suggesting that these genes are derived from the same ancestor. The 6ku3.1.A template was mainly distributed among 10 Cs2ODD-C proteins, except Cs2ODD-C44 and Cs2ODD-C95, and the 5o7y.1.A template was distributed in 9 Cs2ODD-C proteins, except Cs2ODD-C44, Cs2ODD-C21, and Cs2ODD-C121, suggested that the functional differentiation of these 12 *Cs2ODD-C* genes occurred in the process of genome evolution. These results further revealed that these genes played important roles in response to abiotic stress and were also involved in response to different stresses. 

### 2.12. Protein Interaction Networks

The hypothetical protein–protein interaction network was predicted to discover the relationships among different Cs2ODD-C proteins. A total of 32 Cs2ODD-Cs were involved in the interaction networks, which were part of various protein–protein interaction networks (Figure 11). As shown in Figure 11, two Cs2ODD-C proteins (Cs2ODD-C32 and Cs2ODD-C44) interact with two *Arabidopsis* LDOX homologs in *C. sinensis* (Cs2ODD-C57 and Cs2ODD-C63). Moreover, two GA2OX8 proteins in *C. sinensis* (Cs2ODD-C34 and Cs2ODD-C59) interact with six Cs2ODD-C proteins (Cs2ODD-C29, Cs2ODD-C30, Cs2ODD-C31, Cs2ODD-C37, Cs2ODD-C38, and Cs2ODD-C60), and Cs2ODD-C149 interacts with five Cs2ODD-C proteins (Cs2ODD-C26, Cs2ODD-C44, Cs2ODD-C48, Cs2ODD-C49, and Cs2ODD-C50). The interaction network further showed the cascade interactions among five Cs2ODD-C proteins. Specifically, Cs2ODD-C60 interacts with Cs2ODD-C146, Cs2ODD-C146 interacts with Cs2ODD-C145, and Cs2ODD-C145 interacts with Cs2ODD-C72 and Cs2ODD-C95, respectively (Figure 11; Appendix A).

## 3. Discussion 

Genes encoding members of the 2OGD superfamily, as the second largest enzyme family in plants, play important roles in growth and development [6,7,30], including proline hydroxylation, the biosynthesis of secondary metabolites, DNA demethylation, and others [16,31,32,33,34,35]. The ODD-C family of genes, accounting for the majority of *2OGD* genes in plants, plays important roles in the biosynthesis or degradation of various secondary metabolites, including hormones, flavonoids, glucosinolate, benzoxazinoid, and monoterpenoid indole alkaloid [6]. However, a systematic characterization of *Cs2ODD-C* genes in *C. sinensis* has not yet been performed. In this study, the genome-wide identification and characterization of Cs2ODD-C family genes in *C. sinensis* were carried out. A total of 153 Cs2ODD-C genes were identified and divided into 21 groups based on phylogeny, gene structure, and protein motif analyses. The number of Cs2ODD-C family genes in *C. sinensis* was less than in *Glycine max* (209) and *Brassica rapa* (154), but more than in *Zea mays* (75), *O. sativa* (78), *Vitis vinifera* (103), *Arabidopsis* (93), and *Fragaria vesca* (123). These results showed that the species-specific duplication events contributed to the expansion of the Cs2ODD-C gene family in *C. sinensis* [36,37].

Gene duplication makes the primary contribution to gene family expansion and genetic novelty. Several patterns of gene duplication, including tandem, proximal, dispersed, and whole-genome duplication (WGD or segmental duplication), contribute differentially to the expansion of specific gene families in plant genomes [38,39,40]. For example, segmental and tandem duplications contributed to the expansion of WRKY and AP2/ERF transcription factor [41,42]. Transposed duplication was responsible for the proliferation of other important gene families, including MADS-box, F-box, and B3 transcription factors in *Brassicales* [43]. In the present study, nearly half of Cs2ODD-C family genes in *C. sinensis* were derived from WGD (or segmental duplication) and tandem duplications, suggesting that these two duplication events played important roles in the expansion of *Cs2ODD-C* genes in *C. sinensis*. Gene expansion is accompanied by neofunctionalization and subfunctionalization, as well as new protein–protein interactions and gene-expression patterns. For example, *Cs2ODD-C3* and *Cs2ODD-C5* were duplicated and retained from WGD. The *Cs2ODD-C3* gene was highly expressed in leaves and apical buds, whereas *Cs2ODD-C5* showed a high expression level in the roots, flowers, and fruits. Moreover, the expression of *Cs2ODD-C5* was downregulated after MeJA treatment, whereas there was no significant difference in the expression level of *Cs2ODD-C3* under MeJA stress (Figure 6).

In plants, the type of *cis*-acting elements at the 5′ regulatory region (promoter) determines the complex regulatory properties of a given gene [44]. Our result showed that the *cis*-acting elements in the promoters of *Cs2ODD-C* genes, including light-, stress-, and hormone-responsive elements, are involved in light, stress, and hormone responses, and this was consistent with previous studies [45,46,47]. Zhu et al. (2020) identified the *cis*-acting elements in the promoters of key carotenoid pathway genes from Citrus species, which were classified into light-, stress-, and hormone-responsive elements [48]. 

The members of 2ODD-C family are involved in biosynthesis of secondary metabolites, including hormones (auxin, GA, jasmonic acid, salicylic acid, and ethylene) [33,34,35], flavonoids [23], benzylisoquinoline alkaloids [7], glucosinolates [16,31], tropane alkaloids [49], monoterpene indole alkaloids [50], benzoxazinoids [7], coumarins [7], mugineic acid [7], and steroidal glycoalkaloids [12]. These secondary metabolites directly or indirectly respond to abiotic stress. Our results showed that 64.05% (98), 74.5% (114), and 75.16% (115) of Cs2ODD-C family genes in *C. sinensis* showed differential expression patterns under MeJA, PEG, and NaCl treatment, respectively, suggesting that the Cs2ODD-C family genes played essential roles in regulating various abiotic stresses. Moreover, paired comparison analysis further showed that 14, 13, and 49 *Cs2ODD-C* genes displayed the same expression pattern in the MeJA vs. PEG, MeJA vs. NaCl, and PEG vs. NaCl comparisons, implying that these genes might play important roles under these two abiotic stresses, respectively. In addition, two *Cs2ODD-C* genes, *Cs2ODD-C36* and *Cs2ODD-C21*, were up- and downregulated after MeJA, PEG, and NaCl treatments, indicating that these two genes played both positive and negative roles in enhancing the tolerance to abiotic stress. Functional annotation revealed that the orthologous genes of *Cs2ODD-C36* and *Cs2ODD-C21* in *O. sativa* and *Arabidopsis* were *IDS3* (*Os07g07410*) and *GA3ox*, respectively. A previous study revealed that the overexpression of *GA3ox* reduced stress tolerance in *Arabidopsis* [51,52]. Stress-induced DELLA accumulation reduced the bioactive GA content by inhibiting the expression level of *GA3ox*, increased the activities of reactive oxygen species (ROS) detoxification enzymes (catalases and Cu/Zn-superoxide dismutases) and reduced the ROS accumulation in plants [53,54]. In addition, previous studies reported that iron-containing enzymes (superoxide dismutase, catalase, and glutathione peroxidase) were involved in the detoxification of ROS [55,56], and iron deficiency is believed to be dependent on the type and quantity of mugineic acid [57]. A functional annotation showed that the orthologous genes of *Cs2ODD-C36* in *O. sativa* was *IDS3* (Os07g07410) encoding 2′-deoxymugineic-acid 2′-dioxygenase, which was involved in the formation of mugineic acids [58]. Thus, we could propose potential molecular mechanisms for the underlying roles of *Cs2ODD-C* genes in response to various abiotic stresses. One explanation is that stress-induced DELLA accumulation in *C. sinensis* reduces the bioactive GA content by decreasing the expression levels of *Cs2ODD-C21* genes, thus enhancing stress tolerance. Another explanation is that the *Cs2ODD-C36* gene involved in the biosynthesis of mugineic acids plays important roles in enhancing abiotic stress tolerance in *C. sinensis* by improving the absorption of iron to enhance the activity of antioxidant enzymes (superoxide dismutase, catalase, and glutathione peroxidase) (Figure 12).

## 4. Conclusions

In the present study, 153 *Cs2ODD-C* genes were identified in the *C. sinensis* genome and were classified into 21 groups based on the sequence similarity and phylogenetic relationships. The conserved domain, gene structure, and evolutionary relationships of *Cs2ODD-C* genes were also established and analyzed. Investigation of *cis*-regulatory elements of *Cs2ODD-C* genes indicated that many *Cs2ODD-C* genes are involved in regulating abiotic stress tolerance. A comprehensive analysis revealed that two candidate genes, namely *Cs2ODD-C36* and *Cs2ODD-C21*, may be involved in both positively and negatively regulating the multi-stress tolerance. The above results could provide a basis for the functional characterization of *Cs2ODD-C* genes and also provide candidate genes for the future improvement of leaf colorization in *C. sinensis*.

## 5. Materials and Methods

### 5.1. Identification and Classification of Cs2ODD-C Family Genes in C. sinensis

The genome sequence and corresponding annotations of the *C. sinensis* were obtained from the TPIA database (http://tpia.teaplant.org, accessed on 15 February 2022). An HMM file was constructed through the full alignment files of the DIOX_N (PF14226) and 2OG-FeII_Oxy (PF03171) domains, using the hmmbuild program with the HMMER3 software package [59]. The *C. sinensis* protein databases were then used to conduct HMM searches. Short proteins (<100 amino acids) and the redundant sequences mapped to a similar location on the same chromosome were removed from all candidate genes in the chromosomal localizations. Finally, the core domains (DIOX_N and 2OG-FeII_Oxy) were further used to verify the candidate proteins with Pfam (https://pfam.xfam.org/, accessed on 15 February 2022) and SMART (http://smart.embl-heidelberg.de/, accessed on 15 February 2022) [59,60]. Finally, the identified proteins containing DIOX_N and 2OG-Fe II_Oxy domains were regarded as candidate *Cs2ODD-C* genes.

### 5.2. Chromosomal Localization, Motif, and Gene Structure Analyses

The physical location and gene structure of *Cs2ODD-C* genes were acquired in the *C. sinensis* database, and the isodose distribution of the genes was plotted with MapInspect software (http://mapinspect.software.informer.com/, accessed on 15 February 2022). The conserved motifs in *C. sinensis* Cs2ODD-C protein sequences were identified with the online program MEME5.0.1 (http://meme.nbcr.net/meme/intro.html, accessed on 15 February 2022) under the following optimized parameters: maximum motif width, 50 bp; minimum motif width, 6 bp; and maximum numbers of different motifs, 20 [61].

### 5.3. Phylogenetic Analysis

The comparative analysis of full-length 2ODD-C protein sequences between *C. sinensis* and *Arabidopsis* was conducted with Clustal X2.0 software, using default parameters, which included 93 At2ODD-C and 153 Cs2ODD-C protein sequences [6]. The best-fit model of protein evolution was identified with the Model-Generator program [62]. An unrooted phylogenetic tree was constructed based on maximum likelihood (ML) and best model of JTT and G with 100 bootstraps, using PhyML3.0 software. The phylogenetic tree was visualized by using FIGTREE [63].

### 5.4. Synteny Analysis

Synteny analyses between the *Arabidopsis* and *C. sinensis* genomes were performed locally according to the method described by Lee et al. [64]. The potential homologous gene pairs were initially identified across multiple genomes by using BLASTP (E < 1× 10^−5^, top 5 matches). The result of BLASTP was then used as an input file for MCScanX to determine the different type of duplication events (tandem duplication, proximal duplication, WGD/segmental duplication, or dispersed duplication) [65].

### 5.5. Ka/Ks Analysis of Cs2ODD-C Genes

To estimate the evolutionary rates of whole-genome/segmental and tandem-duplicated *Cs2ODD-C* paralog genes, PAL2NAL software was used to construct a multiple-codon alignment from the corresponding aligned protein sequences. Such codon alignments can be used for synonymous (Ks) and nonsynonymous (Ka) estimation [66]. The Ks and Ka substitutions of *Cs2ODD-C* genes were calculated using KaKs_Calculator 2.0, which is commonly used to evaluate the pattern of selection [67]. Generally, Ka/Ks > 1 indicates that genes have undergone positive selection, Ka/Ks < 1 indicates that genes have undergone purifying or negative selection, and Ka/Ks = 1 indicates neutral selection [68].

### 5.6. Expression Analysis 

The expression patterns of *Cs2ODD-C* genes in various organs of *C. sinensis* and under various abiotic stress conditions were obtained from the TPIA database (http://tpia.teaplant.org, accessed on 15 February 2022). A heat map of the reads per kilobase of transcript per million reads mapped (RPKM) values from RNA-seq data for the relative expression levels of Cs2ODD-C family genes was constructed. The inactive genes were defined as those with RPKM < 2 based on previous reports [69,70]. A gene with RPKM > 2 in at least one organ or stress treatment was regarded as a differentially expressed gene in *C. sinensis.*


### 5.7. Growth Conditions, Plant Materials Collection, and Stress Treatments

Tea (*C. sinensis*) seeds were germinated, and the seedlings were planted in a growth chamber (22 °C/18 °C, 14 h photoperiod, sand substrate) for three months. Two-month-old untreated *C. sinensis* seedlings were collected from the growth chamber and used as control samples. Experimental plants were treated with 10 μM MeJA, 10 mM PEG solution, and 200 mM NaCl, respectively. The treated and untreated samples were collected after 24 h and 48 h treatment, immediately frozen in liquid nitrogen, and stored in a deep freezer at −78 °C until further analysis.

### 5.8. RT-qPCR of Cs2ODD-C Genes

Total RNA was extracted from the tea seedling with TaKaRa MiniBEST Plant RNA Extraction Kit (TaKaRA), and the corresponding cDNA was obtained with cDNA Reverse Transcription Kit (TaKaRa). Then qPCR was conducted with SYBR Premix Ex Tag on a DNA Engine Opticon^TM^ 2 system (PCR instrument: Bio-rad T100, manufacturer: Bio-Rad Laboratories Inc, city: State of California, Country: United States). The *CsTBP* (TATA-box binding protein gene)gene was used as a housekeeping gene to normalize the expression level of target genes in *C. sinensis*. All primers used in this study are listed in Appendix A. The reaction conditions of the PCR program were as follows: an initial step at 95 °C for 30 s. After the cycling protocol, melting curves were obtained by increasing the temperature from 60 to 95 °C (0.2 °C^−s^) to denature the double-stranded DNA. Then qPCR amplifications were carried out in 96-well plates. The assays were run in an ABI 7500 system, using the SDS v. 1.4 application software (Applied Biosystems, Foster City, CA, USA). The primer efficiency was created based on a five-fold dilution series of cDNA (1:5, 1:25, 1:50, and 1:100), followed by 42 cycles of 95 °C for 15 s, 58 °C for 15 s, 72 °C for 30 s, and 1 s at 80 °C for reading the plate. The expression levels were evaluated by the 2^−ΔΔCt^ method, with three biological replicates [71].

### 5.9. Analysis of the Tertiary Structures of Candidate Stress-Related Cs2ODD-C Proteins from C. sinensis

The amino acid sequences of 12 candidate stress-related Cs2ODD-C proteins were used as the target sequences to construct the tertiary structures with the SWISS-MODEL program (https://swissmodel.expasy.org, accessed on 15 February 2022). The intensive modeling mode was selected for this analysis.

### 5.10. Prediction of Protein Interaction Networks

The protein interaction networks of Cs2ODD-Cs in *C. sinensis* were predicted using STRING (version 11.0) software (https://www.string-db.org/, accessed on 15 February 2022). The confidence threshold (combined score) was set to 0.5, and “full network” was used as the network type. The protein interaction networks of Cs2ODD-C proteins in *C. sinensis* were visualized using Cytoscape v3.8.2 software [72].

### 5.11. Statistical Analysis

Significant analyses between two sample groups were calculated using the *t*-test analysis [73]. All of the expression analyses were conducted for three biological replicates. The average data of three biological replicates were displayed with plus or minus the standard deviation (average ± SD).

## Figures and Tables

**Figure 1 plants-12-01302-f001:**
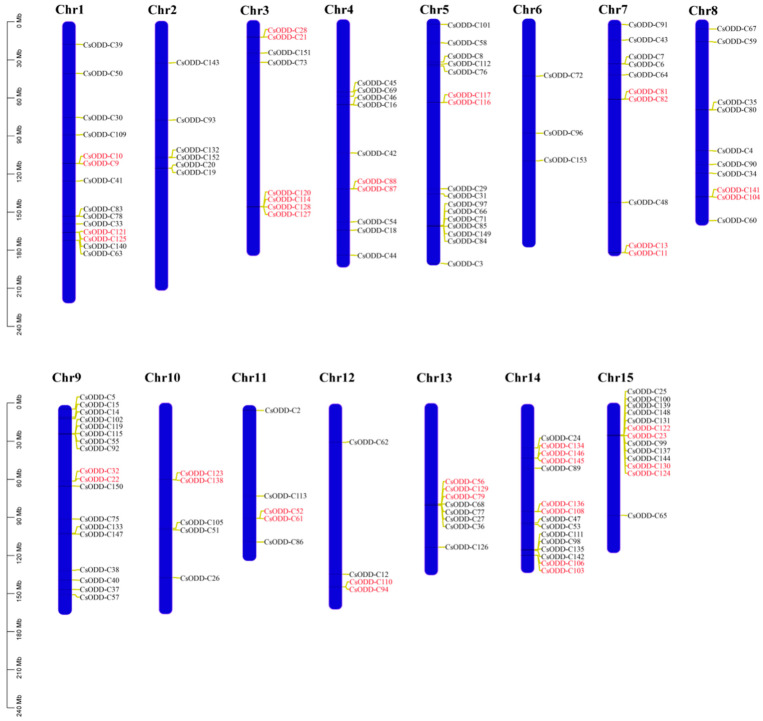
Chromosome distribution and tandem duplication events for *Cs2ODD−C* genes. The position of each *Cs2ODD*−*C* is noted on the right side of each chromosome (Chr). The size of a chromosome is indicated by its relative length. Tandemly duplicated genes are indicated with a red bar.

**Figure 2 plants-12-01302-f002:**
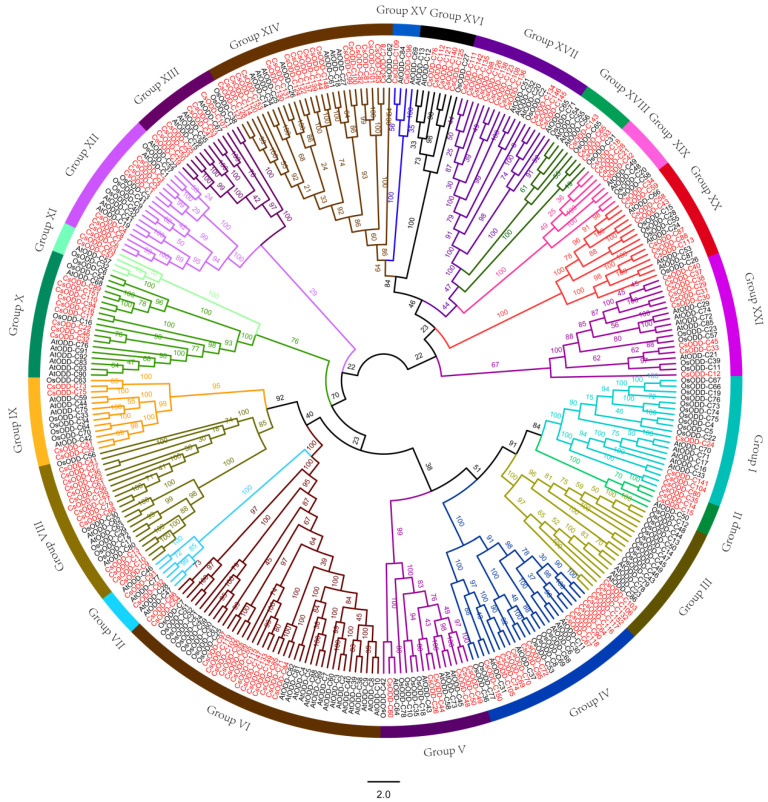
Phylogenetic analysis of 2ODD−C gene family in *C. sinensis*, *O. sativa*, and *Arabidopsis*. An unrooted phylogenetic tree of 2ODD−C gene family among *C. sinensis*, *O. sativa*, and *Arabidopsis* was constructed using the maximum-likelihood method in PhyML 3.0 software, with a bootstrap test (replicated 100 times). The 2ODD−C gene families in *C. sinensis*, *O. sativa*, and *Arabidopsis* are marked red, black, and black, respectively. The values of the bootstrap are shown on the nodes. Different color arcs indicate different groups of ODD−Cs.

**Figure 3 plants-12-01302-f003:**
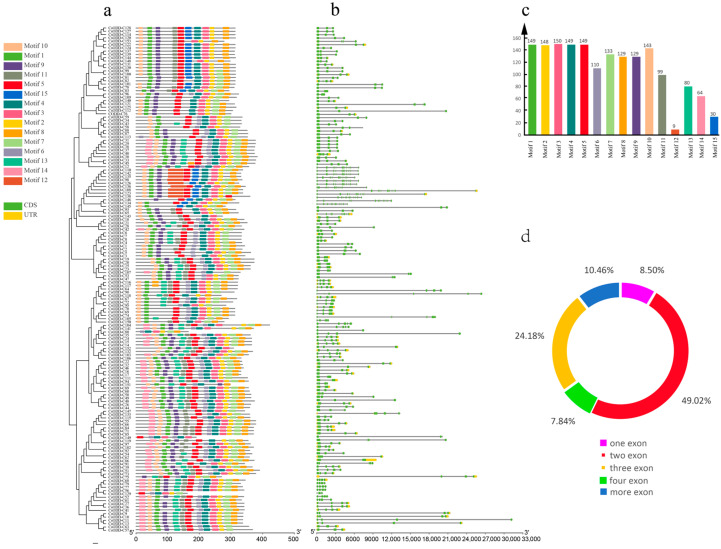
A schematic diagram of conserved motif and gene structure of the Cs2ODD−C gene family. (**a**) The conserved motifs of Cs2ODD−C proteins were conducted using the MEME online program, as described in Section 5. (**b**) The intron and exon structures of the *Cs2ODD−C* genes were obtained by Perl scripts and visualized by GSDS software. (**c**) Statistical analysis of the number of conserved motif distributions in Cs2ODD-C proteins. (**d**) The proportion of the *Cs2ODD−C* genes with different introns.

**Figure 4 plants-12-01302-f004:**
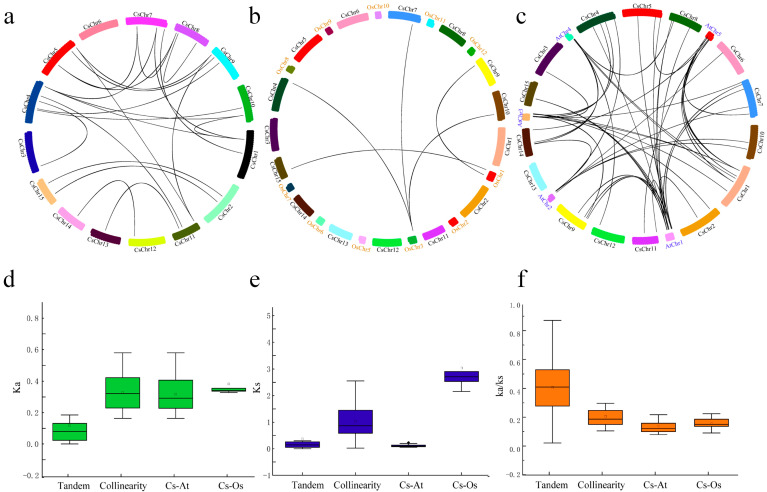
Genomic locations of *Cs2ODD−C* genes and segmentally duplicated gene pairs in the *C. sinensis* genome (**a**) and the orthologous relationships of *Cs2ODD−C* genes with *O. sativa* (**b**) and *Arabidopsis* (**c**). The chromosome number is indicated at the top of each chromosome. Ka (**d**), Ks (**e**), and Ka/Ks (**f**) ratio of segmental duplicate genes and orthologous genes among *C. sinensis*, *O. sativa* and *Arabidopsis*. The box plots exhibit the distributions of Ka, Ks, and Ka/Ks values among paralogs and orthologs. The small square and the line in the box represent average and median values of the Ka, Ks, and Ka/Ks values, respectively. Cs, *C. sinensis*; Os, *O. sativa*; At, *Arabidopsis*.

**Figure 5 plants-12-01302-f005:**
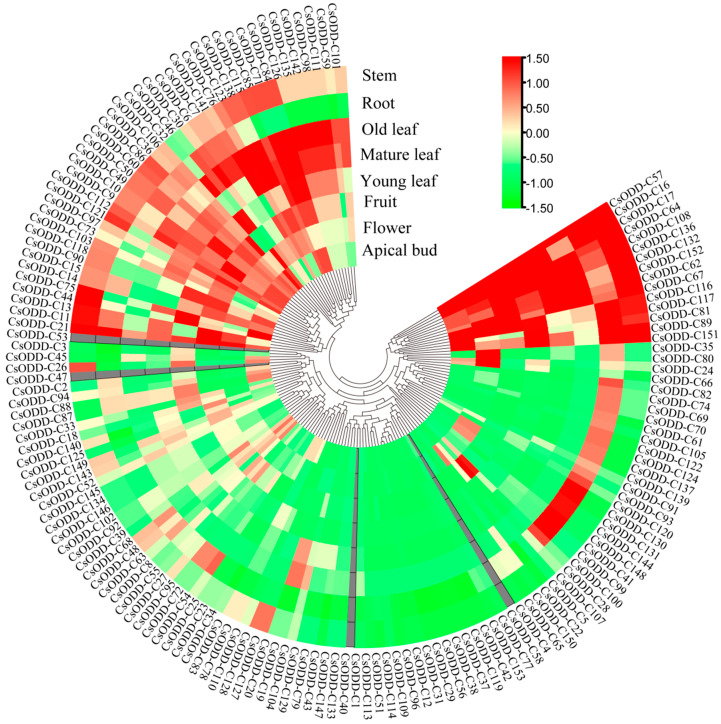
Gene−expression patterns of *Cs2ODD−C* genes in different organs of *C. sinensis* by RNA−seq. The scale bars represent the log2 transformations of the RPKM values. Green and red columns represent the down- and upregulated *Cs2ODD−C* genes, respectively.

**Figure 6 plants-12-01302-f006:**
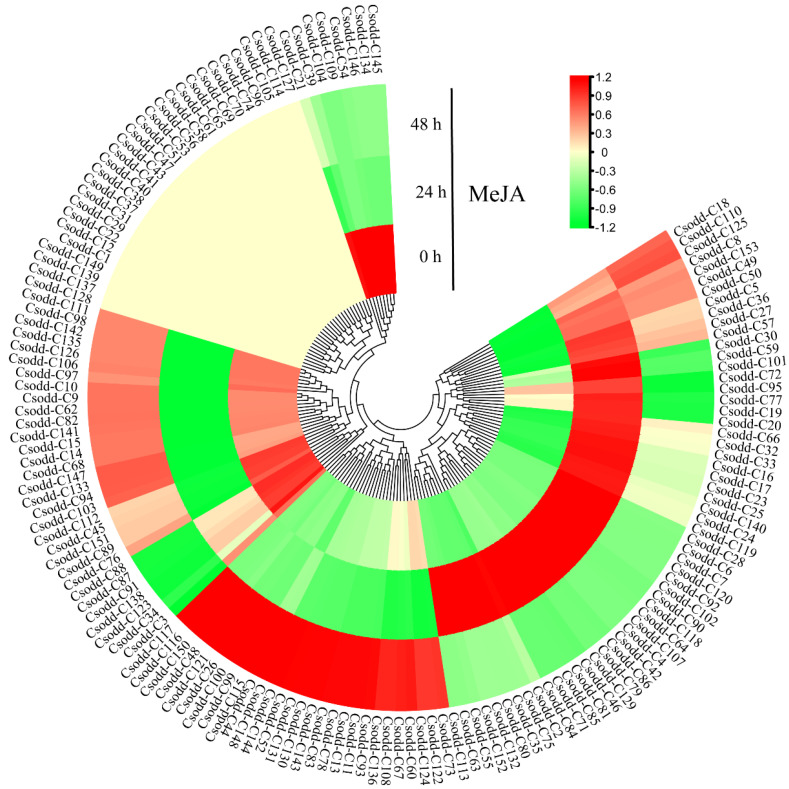
Gene−expression pattern of *Cs2ODD−C* genes in *C. sinensis* after MeJA treatment by RNA−seq. The scale bars represent the log2 transformations of the FPKM values. The colors from red to green indicate the highest-to-lowest log2 (FPKM) values of each gene under the MeJA treatment.

**Figure 7 plants-12-01302-f007:**
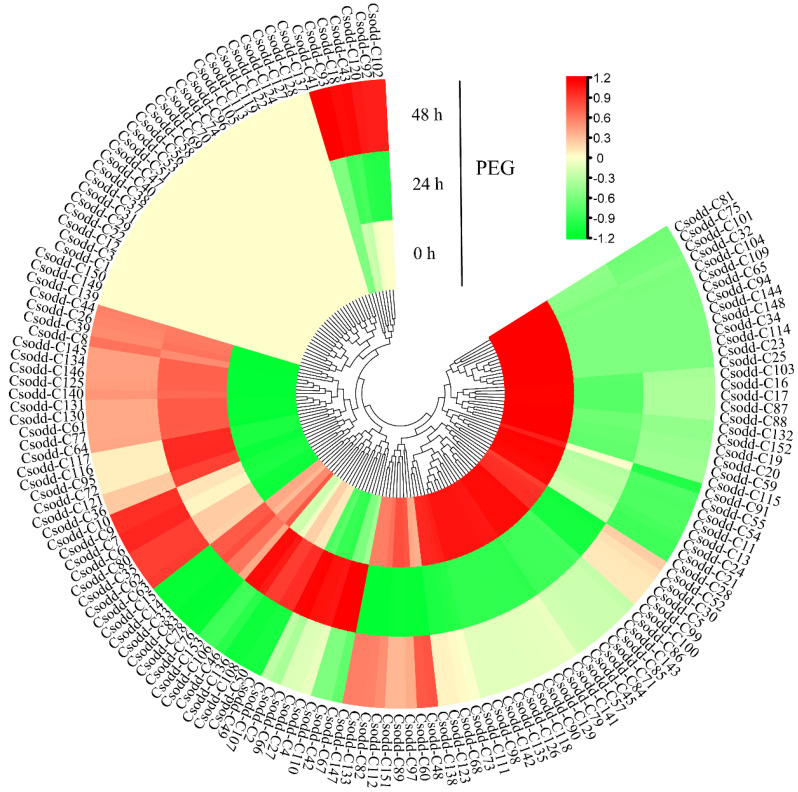
Gene−expression pattern of *Cs2ODD−C* genes in *C. sinensis* after PEG treatment by RNA−seq. The scale bars represent the log2 transformations of the FPKM values. The colors from red to green indicate the highest-to-lowest log2 (FPKM) values of each gene under the PEG treatment.

**Figure 8 plants-12-01302-f008:**
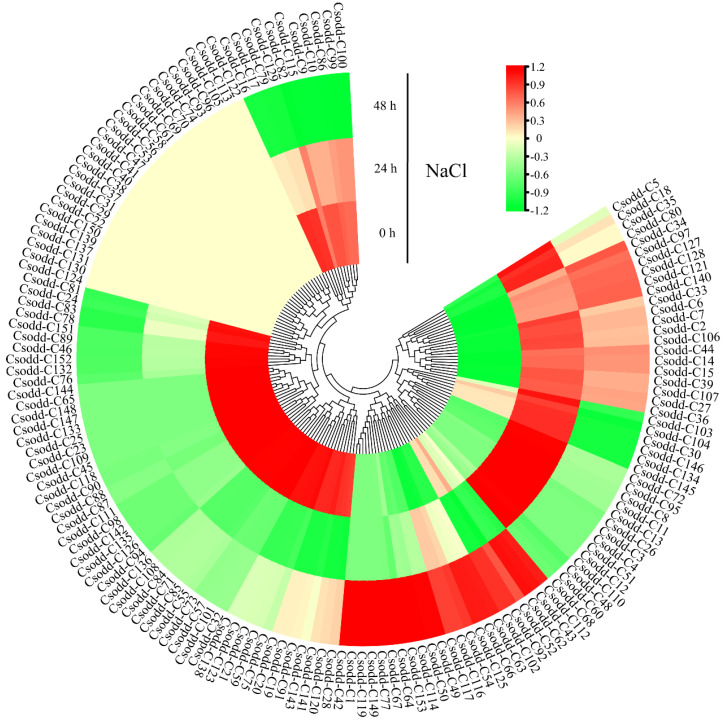
Gene-expression pattern of *Cs2ODD−C* genes in *C. sinensis* after NaCl treatment by RNA−seq. The scale bars represent the log2 transformations of the FPKM values. The colors from red to green indicate the highest-to-lowest log2 (FPKM) values of each gene under the NaCl treatment.

**Figure 9 plants-12-01302-f009:**
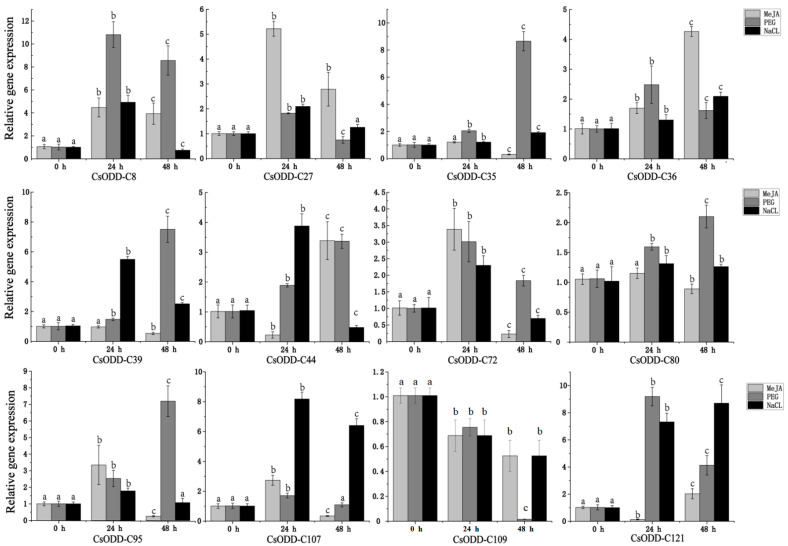
Expression profiles of twelve candidate *Cs2ODD−C* genes in *C. sinensis* under MeJA, PEG, and NaCl treatments. Different lower-case letters indicate a significant difference between the treated (24 h and 48 h) and untreated (0 h) samples (*p* < 0.05).

**Figure 10 plants-12-01302-f010:**
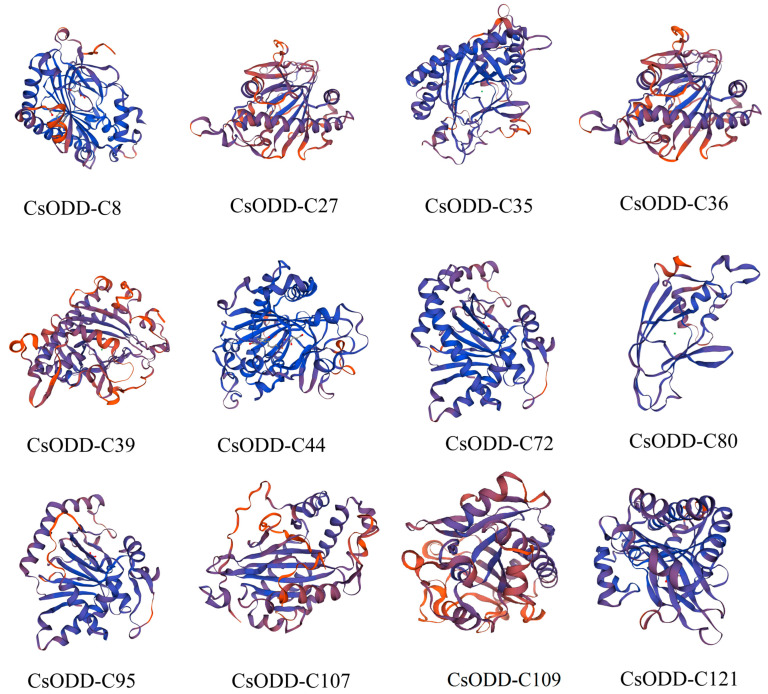
Tertiary structures of 12 candidate Cs2ODD−Cs. The tertiary structure of the protein was analyzed through the online SWISS-MODEL.

**Figure 11 plants-12-01302-f011:**
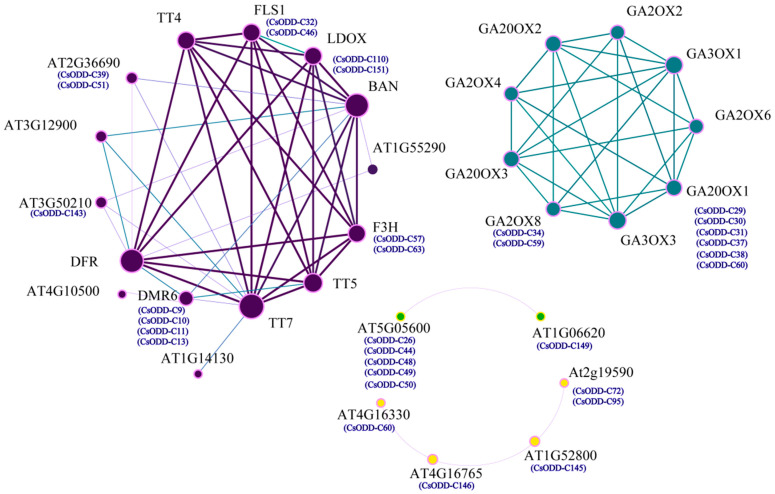
Protein interaction network analysis for the candidate Cs2ODD−Cs involved in response to MeJA, PEG, and NaCl stresses. The online tool STRING was used to predict the network.

**Figure 12 plants-12-01302-f012:**
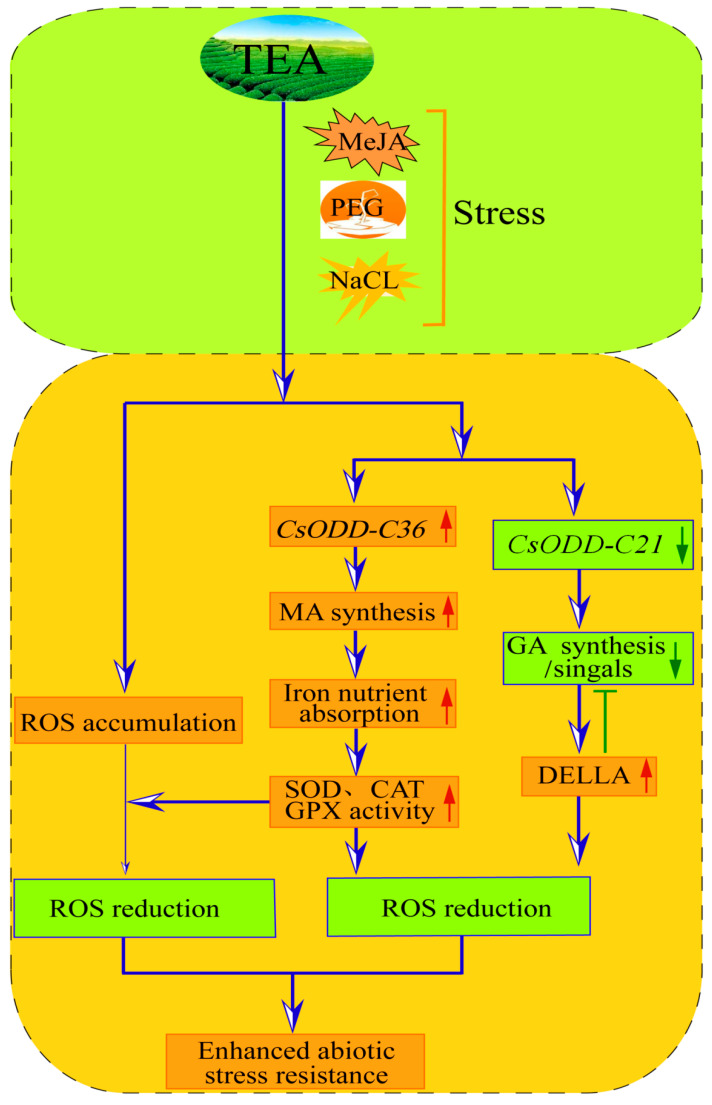
Model describing the *Cs2ODD−C36* and *Cs2ODD−C21* genes involved in regulating the multi-stress tolerance in *C. sinensis*.

**Table 1 plants-12-01302-t001:** The number of the *DOX−C* genes belonging to each group.

Group	No. of *DOX-C Genes*			Clade ^c^	Representative Enzyme
	*C. sinensis*	*A. thaliana*	*O. sativa*		
I	5	5	10	DOXC52	T6OMD, CODM, CJNCSC1
II	2	1	1	DOXC54	SRG
III	4	4	10	DOXC55	–
IV	13	4	6	DOXC53	ACO
V	6	5	5	DOXC46	JAO
VI	15	16	9	DOXC31	D4H, GSLOH, BX6
VII	2	4	0	DOXC30	F6H
VIII	14	3	5	DOXC38	S3H
IX	4	4	5	DOXC37	–
X	9	7	3	DOXC47	FLS, NLS
XI	2	1	1	DOXC28	F3H
XII	8	5	5	DOXC12	C19-GA2ox
XIII	7	4	2	DOXC3	GA3ox
XIV	20	7	1	DOXC20	AOP
XV	2	2	0	DOXC17	–
XVI	5	2	1	DOXC15	DAO
XVII	12	4	2	DOXC24,27	–
XVIII	1	3	1	DOXC23	DIN
XIX	4	2	2	DOXC21	–
XX	7	4	4	DOXC13	C20-GA2ox
XXI	8	4	4	DOXC13	GA2ox
Soloist	3	0	1	–	–

## Data Availability

The datasets presented in this study can be found in online repositories. The names of the repository/repositories and accession number(s) can be found in the article/Appendix A.

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
