# Peer review of "Genome-Wide Identification of 2-Oxoglutarate and Fe (II)-Dependent Dioxygenase (2ODD-C) Family Genes and Expression Profiles under Different Abiotic Stresses in Camellia sinensis (L.)"

_plants, 2023, doi:10.3390/plants12061302_

Round 1

Reviewer 1 Report

In the manuscript titled " Genome-wide identification of 2-oxoglutarate and Fe (II)- de- 2 pendent dioxygenase (2ODD-C) family genes and expression 3 profiles under different abiotic stresses in Camellia sinensis (L.)" the authors carried out the genome wide survey of 2ODD-C gene family in the Camellia sinensis. Authors also carried out the chromosomal distribution, evolutionary, gene structure, protein structure, promoter analyses and interaction analysis of 2ODD-C gene family in the Camellia sinensis. Moreover, the authors have also performed the expression profiling of 2ODD-C genes in Camellia sinensis under tissues and PEG, MeJA, NaCl treatments.

The current manuscript provides a detailed analyses of 2ODD-C gene family in Camellia sinensis. Overall, the study is well designed and performed and include the in-silico analyses to qRT-PCR experiment. However, the manuscript has written in very casual way. In general, the text is written in very poor English and contains numerous ambiguous sentences, spelling errors and grammatical mistakes, which need to be corrected throughout the whole manuscript. Here I just show you a few examples.

1.      Maintain the uniformity in plant names, either write scientific names or common names.

2.      Write the full form of PEG, JA, MeJA and ABA.

3.      Line 71-73, 203-205, 535-537, 540-541: are ambiguous, rewrite the lines

4.      Line 153: Use hyphen symbol instead of “~”.

5.      Moreover, authors should write the gene name in italics throughout the manuscript eg. Line: 180, 183 etc.

6.      Correct the spelling of O. sativa in Table 1.

7.      Authors have shown the five images in figure 3, however in figure legend there are details of only four images, Replace the figure with correct one.

8.      Line 249: Replace the word “suffered” with “experienced”.

9.      Line 255: Authors should use the word “downregulation” in place of “decreased”.

10.  Figure legends of figure 6, 7 and 8 are ambiguous, authors should rewrite them.  

11.  Provide the full form of CsTBA gene, which is used as a reference gene in qRT PCR analysis.

12.  In figure 9, CsODD-C109 graph does not have right hour details, correct it.

13.  Authors should rewrite the tertiary structure analysis results, as present data is not providing any significant information.

14.  Line 506: Non-italicize the word “gene structure”

15.  Line 530: Change the heading and write “Ka/Ks analysis of Cs2ODD-C genes”.

16.  Write the details of statistical test used in the qRT PCR analysis in the materials and methods section.

17.  Provide the information of bootstraps values used in phylogenetic analysis in the Materials and methods heading.

18.  Discussion part lack sufficient information, authors are suggested to discuss the results properly.

19.  Conclusions should be more decisive and not merely repetitions of results.

Author Response

Dear Reviewer 1,

On behalf of my co-authors, we thank you very much for giving us an opportunity to revise our manuscript. We have improved our manuscript based on the comments you and reviewers provided. We appreciate for your warm work earnestly, and hope that the corrections will meet with approval. The responds to the comments are as follows:

1.In the manuscript titled " Genome-wide identification of 2-oxoglutarate and Fe (II)- de- 2 pendent dioxygenase (2ODD-C) family genes and expression 3 profiles under different abiotic stresses in Camellia sinensis (L.)" the authors carried out the genome wide survey of 2ODD-C gene family in the Camellia sinensis. Authors also carried out the chromosomal distribution, evolutionary, gene structure, protein structure, promoter analyses and interaction analysis of 2ODD-C gene family in the Camellia sinensis. Moreover, the authors have also performed the expression profiling of 2ODD-C genes in Camellia sinensis under tissues and PEG, MeJA, NaCl treatments.

The current manuscript provides a detailed analyses of 2ODD-C gene family in Camellia sinensis. Overall, the study is well designed and performed and include the in-silico analyses to qRT-PCR experiment. However, the manuscript has written in very casual way. In general, the text is written in very poor English and contains numerous ambiguous sentences, spelling errors and grammatical mistakes, which need to be corrected throughout the whole manuscript. Here I just show you a few examples.

Response: Thank you for your comments and suggestions. We are sorry that some small errors in this manuscript. The manuscript was edited for correct English language usage, rammar, punctuation and spelling by qualified native English speaking editors at Charlesworth Author Services. Please see:

2.Maintain the uniformity in plant names, either write scientific names or common names.

Response: Thank you for your suggestions and comments. We are sorry that we did not maintain the uniformity in plant names in the manuscript. In the revised manuscript, we have revised the scientific name of plants in MS.

3.   Write the full form of PEG, JA, MeJA and ABA.

Response: Thank you for your suggestions and comments. We are sorry that we did not describe the full form of PEG, JA, MeJA and ABA, in the revised manuscript, we have added the full name of PEG, JA, and MeJA, please see lines 18,72-73.

Lines 18: The expression profiles of Cs2ODD-C genes were explored under methyl jasmonate (MeJA), polyethylene glycol (PEG), and salt (NaCl) stress treatments.

Lines 72-73: In addition, overexpression of the SlF3H gene could improve the cold tolerance of Solanum lycopersicum by regulating jasmonic acid accumulation [19].

  1. Line 71-73, 203-205, 535-537, 540-541: are ambiguous, rewrite the lines

Response: Thank you for your suggestions and comments. We are sorry for the misunderstanding caused by our ignorance, in the revised manuscript, we have rewritten these sentence, please see lines 73-75,204-208,537-541,542-547.

Lines 73-75: Ectopic expression of the DOXC subfamily DoFLS1 gene from Dendrobium officinale promoted flavonol accumulation and enhanced tolerance to abiotic stress in Arabidopsis [20].

Lines 204-208: In addition, the ratio of non-synonymous to synonymous substitutions (Ka/Ks) is a useful measure of the strength and mode of natural selection acting on protein-coding genes. A Ka/Ks ratio of 1 is indicative of neutral selection (no specific direction), lower than 1 indicates purifying selection, and higher than 1 indicates positive selection [28].

Line 537-541: The potential homologous gene pairs were initially identified across multiple genomes by using BLASTP (E < 1e−5, top 5 matches). The result of BLASTP was then used as an input file for MCScanX to determine the different type of duplication events (tandem duplication, proximal duplication, WGD/segmental duplication, or dispersed duplication) [65].

Lines 542-547:

5.5. Ka/Ks analysis of Cs2ODD-C genes

To estimate the evolutionary rates of whole-genome/segmental and tandem-duplicated Cs2ODD-C paralog genes, PAL2NAL software was used to construct a multiple codon alignment from the corresponding aligned protein sequences. Such codon alignments can be used for synonymous (Ks) and nonsynonymous (Ka) estimation [66].

5.Line 153: Use hyphen symbol instead of “~”.

Response: Thank you for your comments. We have replaced “~” with “-”, please see line 156-157.
Line 156-157: The members of groups I–VI and VII–XI commonly contained motifs 11 and 14.

6.Moreover, authors should write the gene name in italics throughout the manuscript eg. Line: 180, 183 etc.

Response: Thank you for your comments. In the revised manuscript, we have rewritten the gene name into italics, please see lines 184-191.

Lines 184-191: The Cs2ODD-C genes contained numerous stress-, hormone-, and light-responsive elements, suggested that these genes might be involved in response to light signalling, plant growth and development, stresses and hormones. In addition, the promoters of Cs2ODD-C36 and Cs2ODD-C21 genes contained many stress-related cis-elements such as CGTCA-motif, TATC-box, GARE-motif, ABRE, MBS, P-box, TCA-motif, and GACG-motif, indicating that these two genes might play crucial roles in response to various abiotic stresses in C. sinensis (Figure S2).

  1. Correct the spelling of O. sativain Table 1.

Response: Thanks for the comment. We have changed “O. sative” to “O. sativa” in Table1 .

8.Authors have shown the five images in figure 3, however in figure legend there are details of only four images, Replace the figure with correct one.

Response: Thank you for your comments. We are sorry for the misunderstanding caused by our ignorance. In the revised manuscript, we have revised Figure 3, please see Figure 3.

  1. Line 249: Replace the word “suffered” with “experienced”.

Response: Thank you for your comments. We have changed “suffered” to “experienced” in line 260.

Line 260: The expression patterns of 153 Cs2ODD-C family genes in C. sinensis seedling experienced MeJA treatment was evaluated used previous RNA-seq data from the TPIA database (Figure 6).

10.Line 255: Authors should use the word “downregulation” in place of “decreased”.

Response: Thank you for your comments. We have changed “downregulation” to “decreased” in Line 267.

Line 267: whereas 14 Cs2ODD-C genes (Cs2ODD-C104, Cs2ODD-C54, Cs2ODD-C91, Cs2ODD-C39, Cs2ODD-C3, Cs2ODD-C87, Cs2ODD-C21, Cs2ODD-C34, Cs2ODD-C123, Cs2ODD-C146, Cs2ODD-C134, Cs2ODD-C145, Cs2ODD-C109, and Cs2ODD-C88) showed downregulation expression at both time points (Figure 6), which may indicate that these are long-term response genes in C. sinensis under MeJA exposure.

11.Figure legends of figure 6, 7 and 8 are ambiguous, authors should rewrite them.

Response: Thank you very much for your valuable comments, in the revised manuscript, we have checked and revised figure legends of figure 6, 7 and 8. Please see figure legends of figure 6, 7 and 8.

Figure legends:

Figure 6. Gene expression pattern of Cs2ODD-C genes in C. sinensis after MeJA treatment by RNA-seq. The scale bars represent the log2 transformations of the FPKM values. The colors from red to green indicated the highest to lowest log2 (FPKM) values of each gene under the MeJA treatment.

Figure 7. Gene expression pattern of Cs2ODD-C genes in C. sinensis after PEG treatment by RNA-seq. The scale bars represent the log2 transformations of the FPKM values. The colors from red to green indicated the highest to lowest log2 (FPKM) values of each gene under the PEG treatment.

Figure 8. Gene expression pattern of Cs2ODD-C genes in C. sinensis after NaCl treatment by RNA-seq. The scale bars represent the log2 transformations of the FPKM values. The colors from red to green indicated the highest to lowest log2 (FPKM) values of each gene under the NaCl treatment.  

12.Provide the full form of CsTBP gene, which is used as a reference gene in qRT PCR analysis.

Response: Thank you very much for your valuable comments, we have provided the full form of CsTBP gene in the revised manuscript, please see Lines 572-573.

Lines 572-573: The CsTBP (TATA-box binding protein) gene was used as a housekeeping gene to normalize the expression level of target genes in C. sinensis.

13.In figure 9, CsODD-C109 graph does not have right hour details, correct it.

Response: Thank you very much for your valuable comments, in the revised manuscript, we have revised the Figure 9, please see Figure 9.

14.Authors should rewrite the tertiary structure analysis results, as present data is not providing any significant information.

Response: Thank you for your suggestions and comments. In the revised manuscript, we have  rewritten the tertiary structure analysis results, please see lines 374-390.

Lines 374-390:

2.11. Tertiary structures of 12 candidate stress-related Cs2ODD-C proteins 

Based on the prediction from SWISS-MODEL (https://swissmodel.expasy.org), the protein tertiary structures of 12 candidate stress-related Cs2ODD-C proteins were constructed according to the C300xA [2ogFe(II) oxygenase family protein] template (Figure 10). The three highest scoring templates were 6lsv.2.A (JOX2), 6ku3.1.A (GA2ox3), and 5o7y.1.A (T6ODM). Seven Cs2ODD-C proteins (Cs2ODD-C27, Cs2ODD-C35, Cs2ODD-C36, Cs2ODD-C39, Cs2ODD-C72, Cs2ODD-C80, and Cs2ODD-C107) contained all three templates (6lsv.2.A, 6ku3.1.A, and 5o7y.1.A), which were used as short- and long-term response genes in C. sinensis under MeJA, PEG and NaCl treatments. The 6lsv.2.A template was present in all 12 Cs2ODD-C proteins, suggesting that these genes are derived from the same ancestor. The 6ku3.1.A template was mainly distributed among 10 Cs2ODD-C proteins except Cs2ODD-C44 and Cs2ODD-C95, and the 5o7y.1.A template was distributed in nine Cs2ODD-C proteins except Cs2ODD-C44, Cs2ODD-C21, and Cs2ODD-C121, suggested that the functional differentiation of these 12 Cs2ODD-C genes occurred in the process of genome evolution. These results further revealed that these genes played important roles in response to abiotic stress, which were also involved in response to different stresses.  

  1. Line 506: Non-italicize the word “gene structure”

Response: Thank you very much for your valuable comments. In the revised manuscript, we have revised the word “gene structure”, please see lines 520.

Lines 520: The physical location and gene structure of Cs2ODD-C genes were acquired in the C. sinensis database, and the isodose distribution of the genes was plotted with MapInspect software (http://mapinspect.software.informer.com/).

16.Line 530: Change the heading and write “Ka/Ks analysis of Cs2ODD-C genes”.

Response: Thank you very much for your valuable comments. We have changed the heading “5.5. Analysis of Ka, Ks, and Ka/Ks values in Cs2ODD-C genes” to  “5.5. Ka/Ks analysis of Cs2ODD-C genes”, please see Line 542.

Lines 542: 5.5. Ka/Ks analysis of Cs2ODD-C genes

17.Write the details of statistical test used in the qRT PCR analysis in the materials and methods section.

Response: Thank you for your valuable comments. We are sorry that we did not describe the statistical test used in the qRT PCR analysis in detail, in the revised manuscript, we describe the details of statistical test used in the qRT PCR analysis, please see lines 596-600.

Lines 596-600:

5.11. Statistical analysis

Significant analysis between two sample groups were calculated using T-test analysis [73]. All of expression analysis were conducted for three biological replicates. The average data of three biological replicates was displayed with plus or minus the standard deviation (average ±SD).

18.Provide the information of bootstraps values used in phylogenetic analysis in the Materials and methods heading.

Response: Thank you very much for your valuable comments. We have checked and added the information of bootstraps values used in phylogenetic analysis in the Materials and methods heading in the revised manuscript, please see Lines 532-534.

Lines 532-534: An unrooted phylogenetic tree was constructed based on maximum likelihood (ML) and best model of JTT and G with 100 bootstraps using PhyML3.0 software. The phylogenetic tree was visualized by using FIGTREE [63].

19.Discussion part lack sufficient information, authors are suggested to discuss the results properly.

Response: Thank you very much for your valuable comments, in the revised manuscript, we have revised the discussion part, please see:

  1. Discussion

Genes encoding members of the 2OGD superfamily, as the second largest enzyme family in plants, play important roles in growth and development [6,7,30], including proline hydroxylation, biosynthesis of secondary metabolites, DNA demethylation, and others [16, 31-35]. The ODD-C family of genes, accounting for the majority of 2OGD genes in plants, play important roles in the biosynthesis or degradation of various secondary metabolites, including hormones, flavonoids, glucosinolate, benzoxazinoid, and monoterpenoid indole alkaloid [6]. However, a systematic characterization of Cs2ODD-C genes in C. sinensis has not been performed. In this study, the genome-wide identification and characterization of Cs2ODD-C family genes in C. sinensis were carried out. A total of 153 Cs2ODD-C genes have been identified and divided into 21 groups based on phylogeny, gene structure and protein motif analyses. The number of Cs2ODD-C family genes in C. sinensis was less than in Glycine max (209) and Brassica rapa (154), but more than in Zea mays (75), O. sativa (78), Vitis vinifera (103), Arabidopsis (93), and Fragaria vesca (123). This results showed that the species-specific duplication events contributed to the expansion of Cs2ODD-C gene family in C. sinensis [36,37].

Gene duplication has primary contribution to gene family expansion and genetic novelty. Several patterns of gene duplication, including tandem, proximal, dispersed, and whole-genome duplication (WGD or segmental duplication), contribute differentially to the expansion of specific gene families in plant genomes [38-40]. For example, segmental and tandem duplications contributed to the expansion of WRKY and AP2/ERF transcription factor [41,42]. Transposed duplication was responsible for the proliferation of other important gene families including MADS-box, F-box, B3 transcription factors in Brassicales [43]. In the present study, nearly half of Cs2ODD-C family genes in C. sinensis were derived from WGD (or segmental duplication) and tandem duplications, suggesting that these two duplication events played important roles in the expansion of Cs2ODD-C genes in C. sinensis. Gene expansion is accompanied by neofunctionalization and subfunctionalization, as well as new protein–protein interactions and gene expression patterns. For example, Cs2ODD-C3 and Cs2ODD-C5 were duplicated and retained from WGD. The Cs2ODD-C3 gene was highly expressed in leaves and apical buds, whereas Cs2ODD-C5 showed a high expression level in the roots, flowers, and fruits. Moreover, the expression of Cs2ODD-C5 was down-regulated after MeJA treatment, whereas there was no significant difference in the expression level of Cs2ODD-C3 under MeJA stress (Figure 6).

In plants, the type of cis-acting elements at the 5′ regulatory region (promoter) determines the complex regulatory properties of a given gene [44]. Our result showed that the cis-acting elements in the promoters of Cs2ODD-C genes including light-, stress-, and hormone-responsive elements, are involved in light, stress and hormone responses, which was consistent with previous studies [45-47]. Zhu et al. (2020) identified the cis-acting elements in the promoters of key carotenoid pathway genes from Citrus species, which were classified into light-, stress-, and hormone-responsive elements [48].

The members of 2ODD-C family are involved in biosynthesis of secondary metabolites, including hormones (auxin, GA, jasmonic acid, salicylic acid, and ethylene) [33-35], flavonoids [23], benzylisoquinoline alkaloids [7], glucosinolates [16,31], tropane alkaloids [49], monoterpene indole alkaloids [50], benzoxazinoids [7], coumarins [7], mugineic acid [7], and steroidal glycoalkaloids [12]. These secondary metabolites directly or indirectly respond to abiotic stress. Our results showed that 64.05% (98), 74.5% (114), and 75.16% (115) of Cs2ODD-C family genes in C. sinensis showed differential expression patterns under MeJA, PEG, and NaCl treatment, respectively, suggesting that the Cs2ODD-C family genes played essential roles in regulating various abiotic stresses. Moreover, paired comparison analysis further showed that 14, 13, and 49 Cs2ODD-C genes displayed the same expression pattern in the MeJA vs. PEG, MeJA vs. NaCl, and PEG vs. NaCl comparisons, implying that these genes might play important roles under these two abiotic stresses, respectively. In addition, two Cs2ODD-C genes, Cs2ODD-C36 and Cs2ODD-C21, were up- and down-regulated after MeJA, PEG, and NaCl treatments, indicated that these two genes played positive and negative roles in enhancing the tolerance to abiotic stress. Functional annotation revealed that the orthologous genes of Cs2ODD-C36 and Cs2ODD-C21 in O. sativa and Arabidopsis were IDS3 (Os07g07410) and GA3ox, respectively. Previous study had revealed that overexpression of GA3ox reduced stress tolerance in Arabidopsis [51,52]. Stress-induced DELLA accumulation reduced the bioactive GA content by inhibiting the expression level of GA3ox, increased the activities of reactive oxygen species (ROS) detoxification enzymes (catalases and Cu/Zn-superoxide dismutases) and reduced the  ROS accumulation in plants [53,54]. In addition, previous studies reported that iron-containing enzymes (superoxide dismutase, catalase, and glutathione peroxidase) were involved in the detoxification of ROS [55,56], and iron deficiency is believed to be dependent on the type and quantity of mugineic acid [57]. Functional annotation showed that the orthologous genes of Cs2ODD-C36 in O. sativa was IDS3 (Os07g07410) encoding 2′-deoxymugineic-acid 2′-dioxygenase, which was involved in the formation of mugineic acids [58]. Thus, we could propose potential molecular mechanisms for the underlying the roles of Cs2ODD-C genes in response to various abiotic stresses. One explanation is that stress-induced DELLA accumulation in C. sinensis reduce the bioactive GA content by decreasing the expression levels of Cs2ODD-C21 genes, which enhances stress tolerance. Another explanation is that the Cs2ODD-C36 gene involved in the biosynthesis of mugineic acids plays important roles in enhancing abiotic stress tolerance in C. sinensis by improving the absorption of iron to enhance the activity of antioxidant enzymes (superoxide dismutase, catalase, and glutathione peroxidase) (Figure 12).

20.Conclusions should be more decisive and not merely repetitions of results.

Response: Thank you for your comments and suggestions, in the revised manuscript, we have revised the conclusions of manuscript, please see Lines 494-504.

Lines 494-504:

  1. 4. Conclusions

In the present study, 153 Cs2ODD-C genes were identified in the C. sinensis genome and were classified into 21 groups based on the sequence similarity and phylogenetic relationships. Conserved domain, gene structure, and evolutionary relationships of Cs2ODD-C genes were also established and analyzed. Investigation of cis-regulatory elements of Cs2ODD-C genes indicated that many Cs2ODD-C genes are involved in regulating abiotic stress tolerance. Comprehensive analysis revealed that 2 candidate genes including Cs2ODD-C36 and Cs2ODD-C21 may be involved in positively and negatively regulating the multi-stress tolerance. The above results could provide a basis for the functional characterization of Cs2ODD-C genes, and also provide candidate genes for the future improvement of leaf colorization in C. sinensis.

Reviewer 2 Report

This manuscript entitled as ' Genome-wide identification of 2-oxoglutarate and Fe (II)- dependent dioxygenase (2ODD-C) family genes and expression profiles under different abiotic stresses in Camellia sinensis (L.)' contributes well to our sciences. It brings new knowledge about new genes to improve C. sinensis production and contribute to a wider piece of information about the 2ODD-C genes family whole. However, the authors must work on the scientific writing to improve it further and give attention to the format and styling throughout the manuscript. 

Author Response

  1. This manuscript entitled as ' Genome-wide identification of 2-oxoglutarate and Fe (II)- dependent dioxygenase (2ODD-C) family genes and expression profiles under different abiotic stresses in Camellia sinensis(L.)' contributes well to our sciences. It brings new knowledge about new genes to improve C. sinensis production and contribute to a wider piece of information about the 2ODD-C genes family whole. However, the authors must work on the scientific writing to improve it further and give attention to the format and styling throughout the manuscript.

Response: Thank you for your comments and suggestions. We are sorry that some small errors in this manuscript. The manuscript was edited for correct English language usage, rammar, punctuation and spelling by qualified native English speaking editors at Charlesworth Author Services.

Reviewer 3 Report

this manuscript has been well organized and written, the main concern is the authors should include the reponse of plants to these stresses, such as plant growth parameters and plant physiological parameters. The current version only test the gene expression, which was not enough to justify this conclusion.

Author Response

Dear Reviewer 3,

On behalf of my co-authors, we thank you very much for giving us an opportunity to revise our manuscript. We have improved our manuscript based on the comments you  provided. We appreciate for your warm work earnestly, and hope that the corrections will meet with approval. The responds to the comments are as follows:

Comments and Suggestions for Authors

  1. this manuscript has been well organized and written, the main concern is the authors should include the response of plants to these stresses, such as plant growth parameters and plant physiological parameters. The current version only test the gene expression, which was not enough to justify this conclusion.

Response: Thank you very much for your valuable comments. We are sorry for the misunderstanding caused by our ignorance. The aim of this study is to systematically investigate the Cs2ODD-C family genes for phylogenetic analysis, gene/protein structure analysis, gene duplications and expression patterns in different organs and under different abiotic stress conditions. In addition, two candidate genes (Cs2ODD-C36 and Cs2ODD-C21) were screened to act as multi-stress response genes. Through this study, we hope to develop a novel molecular basis to improve the multi-stress tolerance of plants and promote the development of phytoremediation technology for multi-stress situations.

Author Response

Dear Reviewer 4,

On behalf of my co-authors, we thank you very much for giving us an opportunity to revise our manuscript. We have improved our manuscript based on the comments you  provided. We appreciate for your warm work earnestly, and hope that the corrections will meet with approval. The responds to the comments are as follows:

Comments and Suggestions for Authors

  1. All abbreviations should be first identified before use them even if they were in abstractor another part of the manuscript

Response: Thank you for your suggestions and comments. We are sorry that we did not describe the full name of these scientific name in manuscript, in the revised manuscript, we have added the full name of these scientific name.

  1. The abstract does not report the main findings of the study in a clear manner. Forexample general expressions are used which do not provide useful information to the  Information that is more specific is required in the abstract.

Response: Thank you for your comments. In the revised manuscript, we have revised the abstract of the manuscript, please see lines 10-26.

Lines 10-26: Abstract: The 2-oxoglutarate and Fe (II)-dependent dioxygenase (2ODD-C) family of 2-oxoglutarate-dependent dioxygenases potentially participates in the biosynthesis of various metabolites under various abiotic stresses. However, there is scarce information on the expression profiles and roles of 2ODD-C genes in Camellia sinensis. We identified 153 Cs2ODD-C genes from C. sinensis, which were distributed unevenly on 15 chromosomes. According to the phylogenetic tree topology, these genes were divided into 21 groups distinguished by conserved motifs and intron/exon structure. Gene duplication analyses revealed that 75 Cs2ODD-C genes were expanded and retained after WGD/segmental and tandem duplications. The expression profiles of Cs2ODD-C genes were explored under methyl jasmonate (MeJA), polyethylene glycol (PEG), and salt (NaCl) stress treatments. Expression analysis showed that 14, 13, and 49 Cs2ODD-C genes displayed the same expression pattern under MeJA and PEG treatments, MeJA and NaCl treatments, and PEG and NaCl treatments, respectively. Further analysis showed that two genes, Cs2ODD-C36 and Cs2ODD-C21, were significantly upregulated and downregulated after MeJA, PEG, and NaCl treatments, indicated that these two genes played positive and negative roles in enhancing the multi-stress tolerance. These results provide candidate genes for the use of genetic engineering technology to modify plants by enhancing multi-stress tolerance to promote phytoremediation efficiency.

  1. The introduction does not point out the gap of the literature the study seeks to fill andnovelty of the study over the existing literature. This point showed be further 

Response: hank you for your suggestions and comments. Through comparative transcriptomics, two candidate genes (Cs2ODD-C36 and Cs2ODD-C21) were identified, which may play a positive and negative role in regulating multiple stress tolerance. In the revised manuscript, we have added related content to the introduction of the article, please see lines 83-94.

Lines 83-94: Although the functions of 2ODD-C genes have been extensively investigated in multiple model organisms, these enzymes have not been systematically analyzed in C. sinensis. In our current study, a comprehensive analysis of the Cs2ODD-C gene family in C. sinensis was performed for phylogenetic evolution, gene structure, conserved motifs, chromosome location, gene duplication and expression patterns in different organs and under different abiotic stress conditions. In addition, two candidate genes (Cs2ODD-C36 and Cs2ODD-C21) might play positively and negatively roles in regulating the multi-stress tolerance. Our results provide new insight into the function of Cs2ODD-C genes in C. sinensis and establish a knowledge base for further genetic improvement of C. sinensis. Through this study, we hope to develop a novel molecular basis to improve the multi-stress tolerance of plants and promote the development of phytoremediation technology for multi-stress situations.

  1. Key words are missing

Response: Thank you for your suggestions and comments. In the revised manuscript, we have added the key words, please see lines 28-29:

Lines 28-29: Keywords: C. sinensis, Cs2ODD-C genes, phylogenetic analysis, expression profile, abiotic stresses

  1. The objectives and conclusion of the study are not clear and need to re-write

Response: Thank you for your comments. In the revised manuscript, we have made certain changes to the objectives and conclusion of the study. We identified 153 Cs2ODD-C genes from C. sinensis, According to the phylogenetic tree topology, these genes were divided into 21 groups distinguished by conserved motifs and intron/exon structure. Gene duplication analyses revealed that 75 Cs2ODD-C genes were expanded and retained after WGD/segmental and tandemp duplications. The expression profiles of Cs2ODD-C genes were explored under methyl jasmonate (MeJA), polyethylene glycol (PEG), and salt (NaCl) stress treatments. Fourteen Cs2ODD-C genes displayed the same expression pattern under MeJA and PEG treatments, 13 genes showed similar patterns under MeJA and NaCl treatments, and 49 genes showed similar profiles under PEG and NaCl treatments. Through comparative transcriptomics, two candidate genes (Cs2ODD-C36 and Cs2ODD-C21) were identified, which may play a positive and negative role in regulating multiple stress tolerance. Through this study, we hope to develop a novel molecular basis to improve the multi-stress tolerance of plants and promote the development of phytoremediation technology for multi-stress situations. please see lines 92-94 and 504-514.

Objectives:

Lines 92-94: Through this study, we hope to develop a novel molecular basis to improve the multi-stress tolerance of plants and promote the development of phytoremediation technology for multi-stress situations.

Lines 494-504:

4.Conclusions

In the present study, 153 Cs2ODD-C genes were identified in the C. sinensis genome and were classified into 21 groups based on the sequence similarity and phylogenetic relationships. Conserved domain, gene structure, and evolutionary relationships of Cs2ODD-C genes were also established and analyzed. Investigation of cis-regulatory elements of Cs2ODD-C genes indicated that many Cs2ODD-C genes are involved in regulating abiotic stress tolerance. Comprehensive analysis revealed that 2 candidate genes including Cs2ODD-C36 and Cs2ODD-C21 may be involved in positively and negatively regulating the multi-stress tolerance. The above results could provide a basis for the functional characterization of Cs2ODD-C genes, and also provide candidate genes for the future improvement of leaf colorization in C. sinensis.

  1. A relevant hypothesis for the study is missing from the introduction. A true scientificquestion should be formed

Response: Thank you for your comments and suggestions. In the revised manuscript, we have added the relevant hypothesis for the study in the introduction, please see lines 88-94:

Lines 88-94: In addition, two candidate genes (Cs2ODD-C36 and Cs2ODD-C21) might play positively and negatively roles in regulating the multi-stress tolerance. Our results provide new insight into the function of Cs2ODD-C genes in C. sinensis and establish a knowledge base for further genetic improvement of C. sinensis. Through this study, we hope to develop a novel molecular basis to improve the multi-stress tolerance of plants and promote the development of phytoremediation technology for multi-stress situations.

  1. Simplify the statement in the paper. Please combine and condense the discussion andConclusion

Response: Thank you for your comments and suggestions. In the revised manuscript, we have combined and condensed the discussion and conclusion, please see lines:413-504

Lines 413-504:

  1. Discussion

Genes encoding members of the 2OGD superfamily, as the second largest enzyme family in plants, play important roles in growth and development [6,7,30], including proline hydroxylation, biosynthesis of secondary metabolites, DNA demethylation, and others [16, 31-35]. The ODD-C family of genes, accounting for the majority of 2OGD genes in plants, play important roles in the biosynthesis or degradation of various secondary metabolites, including hormones, flavonoids, glucosinolate, benzoxazinoid, and monoterpenoid indole alkaloid [6]. However, a systematic characterization of Cs2ODD-C genes in C. sinensis has not been performed. In this study, the genome-wide identification and characterization of Cs2ODD-C family genes in C. sinensis were carried out. A total of 153 Cs2ODD-C genes have been identified and divided into 21 groups based on phylogeny, gene structure and protein motif analyses. The number of Cs2ODD-C family genes in C. sinensis was less than in Glycine max (209) and Brassica rapa (154), but more than in Zea mays (75), O. sativa (78), Vitis vinifera (103), Arabidopsis (93), and Fragaria vesca (123). This results showed that the species-specific duplication events contributed to the expansion of Cs2ODD-C gene family in C. sinensis [36,37].

Gene duplication has primary contribution to gene family expansion and genetic novelty. Several patterns of gene duplication, including tandem, proximal, dispersed, and whole-genome duplication (WGD or segmental duplication), contribute differentially to the expansion of specific gene families in plant genomes [38-40]. For example, segmental and tandem duplications contributed to the expansion of WRKY and AP2/ERF transcription factor [41,42]. Transposed duplication was responsible for the proliferation of other important gene families including MADS-box, F-box, B3 transcription factors in Brassicales [43]. In the present study, nearly half of Cs2ODD-C family genes in C. sinensis were derived from WGD (or segmental duplication) and tandem duplications, suggesting that these two duplication events played important roles in the expansion of Cs2ODD-C genes in C. sinensis. Gene expansion is accompanied by neofunctionalization and subfunctionalization, as well as new protein–protein interactions and gene expression patterns. For example, Cs2ODD-C3 and Cs2ODD-C5 were duplicated and retained from WGD. The Cs2ODD-C3 gene was highly expressed in leaves and apical buds, whereas Cs2ODD-C5 showed a high expression level in the roots, flowers, and fruits. Moreover, the expression of Cs2ODD-C5 was down-regulated after MeJA treatment, whereas there was no significant difference in the expression level of Cs2ODD-C3 under MeJA stress (Figure 6).

In plants, the type of cis-acting elements at the 5′ regulatory region (promoter) determines the complex regulatory properties of a given gene [44]. Our result showed that the cis-acting elements in the promoters of Cs2ODD-C genes including light-, stress-, and hormone-responsive elements, are involved in light, stress and hormone responses, which was consistent with previous studies [45-47]. Zhu et al. (2020) identified the cis-acting elements in the promoters of key carotenoid pathway genes from Citrus species, which were classified into light-, stress-, and hormone-responsive elements [48].

The members of 2ODD-C family are involved in biosynthesis of secondary metabolites, including hormones (auxin, GA, jasmonic acid, salicylic acid, and ethylene) [33-35], flavonoids [23], benzylisoquinoline alkaloids [7], glucosinolates [16,31], tropane alkaloids [49], monoterpene indole alkaloids [50], benzoxazinoids [7], coumarins [7], mugineic acid [7], and steroidal glycoalkaloids [12]. These secondary metabolites directly or indirectly respond to abiotic stress. Our results showed that 64.05% (98), 74.5% (114), and 75.16% (115) of Cs2ODD-C family genes in C. sinensis showed differential expression patterns under MeJA, PEG, and NaCl treatment, respectively, suggesting that the Cs2ODD-C family genes played essential roles in regulating various abiotic stresses. Moreover, paired comparison analysis further showed that 14, 13, and 49 Cs2ODD-C genes displayed the same expression pattern in the MeJA vs. PEG, MeJA vs. NaCl, and PEG vs. NaCl comparisons, implying that these genes might play important roles under these two abiotic stresses, respectively. In addition, two Cs2ODD-C genes, Cs2ODD-C36 and Cs2ODD-C21, were up- and down-regulated after MeJA, PEG, and NaCl treatments, indicated that these two genes played positive and negative roles in enhancing the tolerance to abiotic stress. Functional annotation revealed that the orthologous genes of Cs2ODD-C36 and Cs2ODD-C21 in O. sativa and Arabidopsis were IDS3 (Os07g07410) and GA3ox, respectively. Previous study had revealed that overexpression of GA3ox reduced stress tolerance in Arabidopsis [51,52]. Stress-induced DELLA accumulation reduced the bioactive GA content by inhibiting the expression level of GA3ox, increased the activities of reactive oxygen species (ROS) detoxification enzymes (catalases and Cu/Zn-superoxide dismutases) and reduced the  ROS accumulation in plants [53,54]. In addition, previous studies reported that iron-containing enzymes (superoxide dismutase, catalase, and glutathione peroxidase) were involved in the detoxification of ROS [55,56], and iron deficiency is believed to be dependent on the type and quantity of mugineic acid [57]. Functional annotation showed that the orthologous genes of Cs2ODD-C36 in O. sativa was IDS3 (Os07g07410) encoding 2′-deoxymugineic-acid 2′-dioxygenase, which was involved in the formation of mugineic acids [58]. Thus, we could propose potential molecular mechanisms for the underlying the roles of Cs2ODD-C genes in response to various abiotic stresses. One explanation is that stress-induced DELLA accumulation in C. sinensis reduce the bioactive GA content by decreasing the expression levels of Cs2ODD-C21 genes, which enhances stress tolerance. Another explanation is that the Cs2ODD-C36 gene involved in the biosynthesis of mugineic acids plays important roles in enhancing abiotic stress tolerance in C. sinensis by improving the absorption of iron to enhance the activity of antioxidant enzymes (superoxide dismutase, catalase, and glutathione peroxidase) (Figure 12).

  1. Conclusions

In the present study, 153 Cs2ODD-C genes were identified in the C. sinensis genome and were classified into 21 groups based on the sequence similarity and phylogenetic relationships. Conserved domain, gene structure, and evolutionary relationships of Cs2ODD-C genes were also established and analyzed. Investigation of cis-regulatory elements of Cs2ODD-C genes indicated that many Cs2ODD-C genes are involved in regulating abiotic stress tolerance. Comprehensive analysis revealed that 2 candidate genes including Cs2ODD-C36 and Cs2ODD-C21 may be involved in positively and negatively regulating the multi-stress tolerance. The above results could provide a basis for the functional characterization of Cs2ODD-C genes, and also provide candidate genes for the future improvement of leaf colorization in C. sinensis.

Round 2

Reviewer 1 Report

It seems improved now.   best regards santosh

Reviewer 2 Report

The authors did good job in revising their manuscript and i recommend it to be accepted for publication. 

Reviewer 3 Report

the authors did not improve the required concerns from the reviewers.
